# Uncertainty Aware Semi-Supervised Learning on Graph Data

**Xujiang Zhao**[1], **Feng Chen**[1], **Shu Hu**[2], **Jin-Hee Cho**[3]
[1]The University of Texas at Dallas, {xujiang.zhao, feng,chen}@utdallas.edu
[2]University at Buffalo, SUNY, shuhu@buffalo.edu
[3]Virginia Tech, jicho@vt.edu

## Abstract

Thanks to graph neural networks (GNNs), semi-supervised node classification has shown the state-of-the-art performance in graph data. However, GNNs have not considered different types of uncertainties associated with class probabilities to minimize risk of increasing misclassification under uncertainty in real life. In this work, we propose a multi-source uncertainty framework using a GNN that reflects various types of predictive uncertainties in both deep learning and belief/evidence theory domains for node classification predictions. By collecting evidence from the given labels of training nodes, the *Graph-based Kernel Dirichlet distribution Estimation* (GKDE) method is designed for accurately predicting node-level Dirichlet distributions and detecting out-of-distribution (OOD) nodes. We validated the outperformance of our proposed model compared to the state-of-the-art counterparts in terms of misclassification detection and OOD detection based on six real network datasets. We found that dissonance-based detection yielded the best results on misclassification detection while vacuity-based detection was the best for OOD detection. To clarify the reasons behind the results, we provided the theoretical proof that explains the relationships between different types of uncertainties considered in this work.

## 1 Introduction

Inherent uncertainties derived from different root causes have realized as serious hurdles to find effective solutions for real world problems. Critical safety concerns have been brought due to lack of considering diverse causes of uncertainties, resulting in high risk due to misinterpretation of uncertainties (e.g., misdetection or misclassification of an object by an autonomous vehicle). Graph neural networks (GNNs) [12, 21] have received tremendous attention in the data science community. Despite their superior performance in semi-supervised node classification and regression, they didn't consider various types of uncertainties in the their decision process. Predictive uncertainty estimation [11] using Bayesian NNs (BNNs) has been explored for classification prediction and regression in the computer vision applications, based on aleatoric uncertainty (AU) and epistemic uncertainty (EU). AU refers to data uncertainty from statistical randomness (e.g., inherent noises in observations) while EU indicates model uncertainty due to limited knowledge (e.g., ignorance) in collected data. In the belief or evidence theory domain, Subjective Logic (SL) [9] considered vacuity (or a lack of evidence or ignorance) as uncertainty in a subjective opinion. Recently other uncertainty types, such as dissonance, consonance, vagueness, and monosonance [9], have been discussed based on SL to measure them based on their different root causes.

We first considered multidimensional uncertainty types in both deep learning (DL) and belief and evidence theory domains for node-level classification, misclassification detection, and out-of-distribution (OOD) detection tasks. By leveraging the learning capability of GNNs and considering multidimensional uncertainties, we propose a uncertainty-aware estimation framework by quantifying

different uncertainty types associated with the predicted class probabilities. In this work, we made the following **key contributions**:

- **A multi-source uncertainty framework for GNNs**. Our proposed framework first provides the estimation of various types of uncertainty from both DL and evidence/belief theory domains, such as dissonance (derived from conflicting evidence) and vacuity (derived from lack of evidence). In addition, we designed a Graph-based Kernel Dirichlet distribution Estimation (GKDE) method to reduce errors in quantifying predictive uncertainties.
- **Theoretical analysis**: Our work is the first that provides a theoretical analysis about the relationships between different types of uncertainties considered in this work. We demonstrate via a theoretical analysis that an OOD node may have a high predictive uncertainty under GKDE.
- **Comprehensive experiments for validating the performance of our proposed framework**: Based on the six real graph datasets, we compared the performance of our proposed framework with that of other competitive counterparts. We found that the dissonance-based detection yielded the best results in misclassification detection while vacuity-based detection best performed in OOD detection.

Note that we use the term 'predictive uncertainty' in order to mean uncertainty estimated to solve prediction problems.

## 2 Related Work

DL research has mainly considered *aleatoric* uncertainty (AU) and *epistemic* uncertainty (EU) using BNNs for computer vision applications. AU consists of homoscedastic uncertainty (i.e., constant errors for different inputs) and heteroscedastic uncertainty (i.e., different errors for different inputs) [4]. A Bayesian DL framework was presented to simultaneously estimate both AU and EU in regression (e.g., depth regression) and classification (e.g., semantic segmentation) tasks [11]. Later, *distributional uncertainty* was defined based on distributional mismatch between testing and training data distributions [14]. *Dropout variational inference* [5] was used for an approximate inference in BNNs using epistemic uncertainty, similar to *DropEdge* [15]. Other algorithms have considered overall uncertainty in node classification [3, 13, 22]. However, no prior work has considered uncertainty decomposition in GNNs.

In the belief (or evidence) theory domain, uncertainty reasoning has been substantially explored, such as Fuzzy Logic [1], Dempster-Shafer Theory (DST) [19], or Subjective Logic (SL) [8]. Belief theory focuses on reasoning inherent uncertainty in information caused by unreliable, incomplete, deceptive, or conflicting evidence. SL considered predictive uncertainty in subjective opinions in terms of *vacuity* (i.e., a lack of evidence) and *vagueness* (i.e., failing in discriminating a belief state) [8]. Recently, other uncertainty types have been studied, such as *dissonance* caused by conflicting evidence[9]. In the deep NNs, [18] proposed evidential deep learning (EDL) model, using SL to train a deterministic NN for supervised classification in computer vision based on the sum of squared loss. However, EDL didn't consider a general method of estimating multidimensional uncertainty or graph structure.

## 3 Multidimensional Uncertainty and Subjective Logic

This section provides an overview of SL and discusses multiple types of uncertainties estimated based on SL, called *evidential uncertainty*, with the measures of *vacuity* and *dissonance*. In addition, we give a brief overview of *probabilistic uncertainty*, discussing the measures of *aleatoric* uncertainty and *epistemic* uncertainty.

### 3.1 Subjective Logic

A multinomial opinion of a random variable $y$ is represented by $\omega = (\boldsymbol{b}, u, \boldsymbol{a})$ where a domain is $\mathbb{Y} \equiv \{1, \cdots, K\}$ and the additivity requirement of $\omega$ is given as $\sum_{k \in \mathbb{Y}} b_k + u = 1$. To be specific, each parameter indicates,

- $\boldsymbol{b}$: *belief mass distribution* over $\mathbb{Y}$ and $\boldsymbol{b} = [b_1, \ldots, b_K]^T$;
- $u$: *uncertainty mass* representing *vacuity of evidence*;
- $\boldsymbol{a}$: *base rate distribution* over $\mathbb{Y}$ and $\boldsymbol{a} = [a_1, \ldots, a_K]^T$.

The projected probability distribution of a multinomial opinion can be calculated as:

$$P(y = k) = b_k + a_k u, \quad \forall k \in \mathbb{Y}. \tag{1}$$

A multinomial opinion $\omega$ defined above can be equivalently represented by a $K$-dimensional Dirichlet probability density function (PDF), where the special case with $K = 2$ is the Beta PDF as a binomial opinion. Let $\boldsymbol{\alpha}$ be a strength vector over the singletons (or classes) in $\mathbb{Y}$ and $\mathbf{p} = [p_1, \cdots, p_K]^T$ be a probability distribution over $\mathbb{Y}$. The Dirichlet PDF with $\mathbf{p}$ as a random vector $K$-dimensional variables is defined by:

$$\mathrm{Dir}(\boldsymbol{p}|\boldsymbol{\alpha}) = \frac{1}{B(\boldsymbol{\alpha})} \prod\nolimits_{k \in \mathbb{Y}} p_k^{(\alpha_k - 1)}, \tag{2}$$

where $\frac{1}{B(\boldsymbol{\alpha})} = \frac{\Gamma(\sum_{k \in \mathbb{Y}} \alpha_k)}{\prod_{k \in \mathbb{Y}} (\alpha_k)}$, $\alpha_k \geq 0$, and $p_k \neq 0$, if $\alpha_k < 1$.

The term *evidence* is introduced as a measure of the amount of supporting observations collected from data that a sample should be classified into a certain class. Let $e_k$ be the evidence derived for the class $k \in \mathbb{Y}$. The total strength $\alpha_k$ for the belief of each class $k \in \mathbb{Y}$ can be calculated as: $\alpha_k = e_k + a_k W$, where $e_k \geq 0, \forall k \in \mathbb{Y}$, and $W$ refers to a non-informative weight representing the amount of uncertain evidence. Given the Dirichlet PDF as defined above, the expected probability distribution over $\mathbb{Y}$ can be calculated as:

$$\mathbb{E}[p_k] = \frac{\alpha_k}{\sum_{k=1}^{K} \alpha_k} = \frac{e_k + a_k W}{W + \sum_{k=1}^{K} e_k}. \tag{3}$$

The observed evidence in a Dirichlet PDF can be mapped to a multinomial opinion as follows:

$$b_k = \frac{e_k}{S}, \; u = \frac{W}{S}, \tag{4}$$

where $S = \sum_{k=1}^{K} \alpha_k$ refers to the Dirichlet strength. Without loss of generality, we set $a_k = \frac{1}{K}$ and the non-informative prior weight (i.e., $W = K$), which indicates that $a_k \cdot W = 1$ for each $k \in \mathbb{Y}$.

### 3.2 Evidential Uncertainty

In [9], we discussed a number of multidimensional uncertainty dimensions of a subjective opinion based on the formalism of SL, such as singularity, vagueness, vacuity, dissonance, consonance, and monosonance. These uncertainty dimensions can be observed from binomial, multinomial, or hyper opinions depending on their characteristics (e.g., the vagueness uncertainty is only observed in hyper opinions to deal with composite beliefs). In this paper, we discuss two main uncertainty types that can be estimated in a multinomial opinion, which are *vacuity* and *dissonance*.

The main cause of vacuity is derived from a lack of evidence or knowledge, which corresponds to the uncertainty mass, $u$, of a multinomial opinion in SL as: $vac(\omega) \equiv u = K/S$, as estimated in Eq. (4). This uncertainty exists because the analyst may have insufficient information or knowledge to analyze the uncertainty. The *dissonance* of a multinomial opinion can be derived from the same amount of conflicting evidence and can be estimated based on the difference between singleton belief masses (e.g., class labels), which leads to 'inconclusiveness' in decision making applications. For example, a four-state multinomial opinion is given as $(b_1, b_2, b_3, b_4, u, a) = (0.25, 0.25, 0.25, 0.25, 0.0, a)$ based on Eq. (4), although the vacuity $u$ is zero, a decision can not be made if there are the same amounts of beliefs supporting respective beliefs. Given a multinomial opinion with non-zero belief masses, the measure of dissonance can be calculated as:

$$diss(\omega) = \sum_{i=1}^{K} \Big( \frac{b_i \sum_{j \neq i} b_j \mathrm{Bal}(b_j, b_i)}{\sum_{j \neq i} b_j} \Big), \tag{5}$$

where the relative mass balance between a pair of belief masses $b_j$ and $b_i$ is defined as $\mathrm{Bal}(b_j, b_i) = 1 - |b_j - b_i|/(b_j + b_i)$. We note that the dissonance is measured only when the belief mass is non-zero. If all belief masses equal to zero with vacuity being 1 (i.e., $u = 1$), the dissonance will be set to zero.

### 3.3 Probabilistic Uncertainty

For classification, the estimation of the probabilistic uncertainty relies on the design of an appropriate Bayesian DL model with parameters $\boldsymbol{\theta}$. Given input $x$ and dataset $\mathcal{G}$, we estimate a class probability by $P(y|x) = \int P(y|x; \boldsymbol{\theta}) P(\boldsymbol{\theta}|\mathcal{G}) d\boldsymbol{\theta}$, and obtain ***epistemic uncertainty*** estimated by mutual information [2, 14]:

$$\underbrace{I(y, \boldsymbol{\theta}|x, \mathcal{G})}_{Epistemic} = \underbrace{\mathcal{H}\big[\mathbb{E}_{P(\boldsymbol{\theta}|\mathcal{G})}[P(y|x; \boldsymbol{\theta})]\big]}_{Entropy} - \underbrace{\mathbb{E}_{P(\boldsymbol{\theta}|\mathcal{G})}\big[\mathcal{H}[P(y|x; \boldsymbol{\theta})]\big]}_{Aleatoric}, \tag{6}$$

where $\mathcal{H}(\cdot)$ is Shannon's entropy of a probability distribution. The first term indicates **entropy** that represents the total uncertainty while the second term is **aleatoric** that indicates data uncertainty. By computing the difference between entropy and aleatoric uncertainties, we obtain epistemic uncertainty, which refers to uncertainty from model parameters.

## 4 Relationships Between Multiple Uncertainties

We use the shorthand notations $u_v$, $u_{diss}$, $u_{alea}$, $u_{epis}$, and $u_{en}$ to represent vacuity, dissonance, aleatoric, epistemic, and entropy, respectively.

To interpret multiple types of uncertainty, we show three prediction scenarios of 3-class classification in Figure 1, in each of which the strength parameters $\alpha = [\alpha_1, \alpha_2, \alpha_3]$ are known. To make a prediction with high confidence, the subjective multinomial opinion, following a Dirichlet distribution, will yield a sharp distribution on one corner of the simplex (see Figure 1 (a)). For a prediction with conflicting evidence, called a conflicting prediction (CP), the multinomial opinion should yield a central distribution, representing confidence to predict a flat categorical distribution over class labels (see Figure 1 (b)). For an OOD scenario with $\alpha = [1, 1, 1]$, the multinomial opinion would yield a flat distribution over the simplex (Figure 1 (c)), indicating high uncertainty due to the lack of evidence. The first technical contribution of this work is as follows.

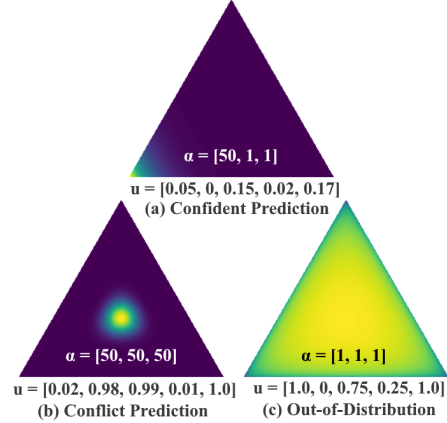

Figure 1: Multiple uncertainties of different prediction. Let $\mathbf{u} = [u_v, u_{diss}, u_{alea}, u_{epis}, u_{en}]$.

**Theorem 1** *We consider a simplified scenario, where a multinomial random variable $y$ follows a K-class categorical distribution: $y \sim Cal(\boldsymbol{p})$, the class probabilities $\boldsymbol{p}$ follow a Dirichlet distribution: $\boldsymbol{p} \sim Dir(\boldsymbol{\alpha})$, and $\boldsymbol{\alpha}$ refer to the Dirichlet parameters. Given a total Dirichlet strength $S = \sum_{i=1}^{K} \alpha_i$, for any opinion $\omega$ on a multinomial random variable $y$, we have*

1. *General relations on all prediction scenarios.*

    *(a) $u_v + u_{diss} \leq 1$; (b) $u_v > u_{epis}$.*

2. *Special relations on the OOD and the CP.*

    *(a) For an OOD sample with a uniform prediction (i.e., $\alpha = [1, \ldots, 1]$), we have*
    $$1 = u_v = u_{en} > u_{alea} > u_{epis} > u_{diss} = 0$$

    *(b) For an in-distribution sample with a conflicting prediction (i.e., $\alpha = [\alpha_1, \ldots, \alpha_K]$ with $\alpha_1 = \alpha_2 = \cdots = \alpha_K$, if $S \to \infty$), we have*
    $$u_{en} = 1, \lim_{S \to \infty} u_{diss} = \lim_{S \to \infty} u_{alea} = 1, \lim_{S \to \infty} u_v = \lim_{S \to \infty} u_{epis} = 0$$
    *with $u_{en} > u_{alea} > u_{diss} > u_v > u_{epis}$.*

The proof of Theorem 1 can be found in Appendix A.1. As demonstrated in Theorem 1 and Figure 1, entropy cannot distinguish OOD (see Figure 1 (c)) and conflicting predictions (see Figure 1 (b)) because entropy is high for both cases. Similarly, neither aleatoric uncertainty nor epistemic uncertainty can distinguish OOD from conflicting predictions. In both cases, aleatoric uncertainty is high while epistemic uncertainty is low. On the other hand, vacuity and dissonance can clearly distinguish OOD from a conflicting prediction. For example, OOD objects typically show high vacuity with low dissonance while conflicting predictions exhibit low vacuity with high dissonance. This observation is confirmed through the empirical validation via our extensive experiments in terms of misclassification and OOD detection tasks.

## 5 Uncertainty-Aware Semi-Supervised Learning

In this section, we describe our proposed uncertainty framework based on semi-supervised node classification problem. It is designed to predict the subjective opinions about the classification

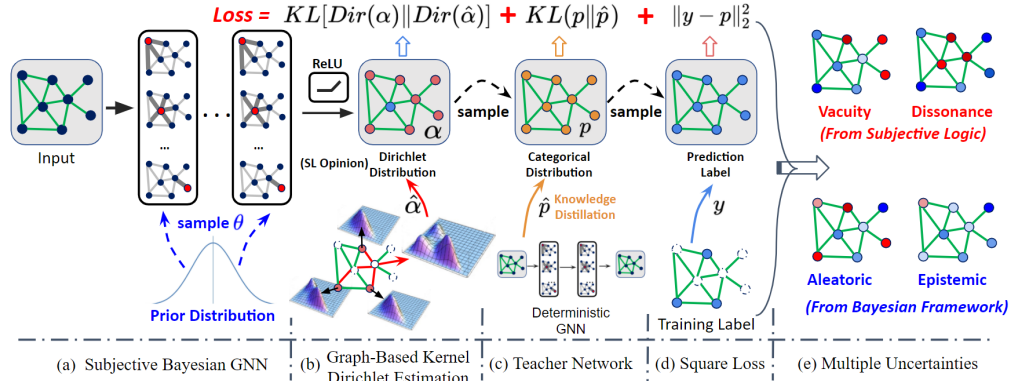

Figure 2: **Uncertainty Framework Overview.** Subjective Bayesian GNN (a) designed for estimating the different types of uncertainties. The loss function includes square error (d) to reduce bias, GKDE (b) to reduce errors in uncertainty estimation and teacher network (c) to refine class probability.

of testing nodes, such that a variety of uncertainty types, such as vacuity, dissonance, aleatoric uncertainty, and epistemic uncertainty, can be quantified based on the estimated subjective opinions and posterior of model parameters. As a subjective opinion can be equivalently represented by a Dirichlet distribution about the class probabilities, we proposed a way to predict the node-level subjective opinions in the form of node-level Dirichlet distributions. The overall description of the framework is shown in Figure 2.

## 5.1 Problem Definition

Given an input graph $\mathcal{G} = (\mathbb{V}, \mathbb{E}, \mathbf{r}, \mathbf{y}_{\mathbb{L}})$, where $\mathbb{V} = \{1, \ldots, N\}$ is a ground set of nodes, $\mathbb{E} \subseteq \mathbb{V} \times \mathbb{V}$ is a ground set of edges, $\mathbf{r} = [\mathbf{r}_1, \cdots, \mathbf{r}_N]^T \in \mathbb{R}^{N \times d}$ is a node-level feature matrix, $\mathbf{r}_i \in \mathbb{R}^d$ is the feature vector of node $i$, $\mathbf{y}_{\mathbb{L}} = \{y_i \mid i \in \mathbb{L}\}$ are the labels of the training nodes $\mathbb{L} \subset \mathbb{V}$, and $y_i \in \{1, \ldots, K\}$ is the class label of node $i$. **We aim to predict**: (1) the **class probabilities** of the testing nodes: $\mathbf{p}_{\mathbb{V} \setminus \mathbb{L}} = \{\mathbf{p}_i \in [0,1]^K \mid i \in \mathbb{V} \setminus \mathbb{L}\}$; and (2) the **associated multidimensional uncertainty estimates** introduced by different root causes: $\mathbf{u}_{\mathbb{V} \setminus \mathbb{L}} = \{\mathbf{u}_i \in [0,1]^m \mid i \in \mathbb{V} \setminus \mathbb{L}\}$, where $p_{i,k}$ is the probability that the class label $y_i = k$ and $m$ is the total number of uncertainty types.

## 5.2 Proposed Uncertainty Framework

**Learning evidential uncertainty.** As discussed in Section 3.1, evidential uncertainty can be derived from multinomial opinions or equivalently Dirichlet distributions to model a probability distribution for the class probabilities. Therefore, we design a Subjective GNN (S-GNN) $f$ to form their multinomial opinions for the node-level Dirichlet distribution $\text{Dir}(\mathbf{p}_i|\boldsymbol{\alpha}_i)$ of a given node $i$. Then, the conditional probability $P(\mathbf{p}|A, \mathbf{r}; \boldsymbol{\theta})$ can be obtained by:

$$P(\mathbf{p}|A, \mathbf{r}; \boldsymbol{\theta}) = \prod_{i=1}^{N} \text{Dir}(\mathbf{p}_i|\boldsymbol{\alpha}_i), \ \boldsymbol{\alpha}_i = f_i(A, \mathbf{r}; \boldsymbol{\theta}), \tag{7}$$

where $f_i$ is the output of S-GNN for node $i$, $\boldsymbol{\theta}$ is the model parameters, and $A$ is an adjacency matrix. The Dirichlet probability function $\text{Dir}(\mathbf{p}_i|\boldsymbol{\alpha}_i)$ is defined by Eq. (2).

Note that S-GNN is similar to classical GNN, except that we use an activation layer (e.g., *ReLU*) instead of the *softmax* layer (only outputs class probabilities). This ensures that S-GNN would output non-negative values, which are taken as the parameters for the predicted Dirichlet distribution.

**Learning probabilistic uncertainty.** Since probabilistic uncertainty relies on a Bayesian framework, we proposed a Subjective Bayesian GNN (S-BGNN) that adapts S-GNN to a Bayesian framework, with the model parameters $\boldsymbol{\theta}$ following a prior distribution. The joint class probability of $\mathbf{y}$ can be estimated by:

$$P(\mathbf{y}|A, \mathbf{r}; \mathcal{G}) = \int \int P(\mathbf{y}|\mathbf{p}) P(\mathbf{p}|A, \mathbf{r}; \boldsymbol{\theta}) P(\boldsymbol{\theta}|\mathcal{G}) d\mathbf{p} d\boldsymbol{\theta}$$

$$\approx \frac{1}{M} \sum_{m=1}^{M} \sum_{i=1}^{N} \int P(\mathbf{y}_i|\mathbf{p}_i) P(\mathbf{p}_i|A, \mathbf{r}; \boldsymbol{\theta}^{(m)}) d\mathbf{p}_i, \quad \boldsymbol{\theta}^{(m)} \sim q(\boldsymbol{\theta}) \tag{8}$$

where $P(\boldsymbol{\theta}|\mathcal{G})$ is the posterior, estimated via dropout inference, that provides an approximate solution of posterior $q(\boldsymbol{\theta})$ and taking samples from the posterior distribution of models [5]. Thanks to the

benefit of dropout inference, training a DL model directly by minimizing the cross entropy (or square error) loss function can effectively minimize the KL-divergence between the approximated distribution and the full posterior (i.e., $\text{KL}[q(\boldsymbol{\theta})\|P(\theta|\mathcal{G})]$) in variational inference [5, 10]. For interested readers, please refer to more detail in Appendix B.8.

Therefore, training S-GNN with stochastic gradient descent enables learning of an approximated distribution of weights, which can provide good explainability of data and prevent overfitting. We use a *loss function* to compute its Bayes risk with respect to the sum of squares loss $\|\mathbf{y} - \mathbf{p}\|_2^2$ by:

$$\mathcal{L}(\boldsymbol{\theta}) = \sum_{i \in \mathbb{L}} \int \|\mathbf{y}_i - \mathbf{p}_i\|_2^2 \cdot P(\mathbf{p}_i | A, \mathbf{r}; \boldsymbol{\theta}) d\mathbf{p}_i = \sum_{i \in \mathbb{L}} \sum_{k=1}^{K} (y_{ik} - \mathbb{E}[p_{ik}])^2 + \text{Var}(p_{ik}), \quad (9)$$

where $\mathbf{y}_i$ is an one-hot vector encoding the ground-truth class with $y_{ij} = 1$ and $y_{ik} \neq$ for all $k \neq j$ and $j$ is a class label. Eq. (9) aims to minimize the prediction error and variance, leading to maximizing the classification accuracy of each training node by removing excessive misleading evidence.

### 5.3 Graph-based Kernel Dirichlet distribution Estimation (GKDE)

The loss function in Eq. (9) is designed to measure the sum of squared loss based on class labels of training nodes. However, it does not directly measure the quality of the predicted node-level Dirichlet distributions. To address this limitation, we proposed *Graph-based Kernel Dirichlet distribution Estimation* (GKDE) to better estimate node-level Dirichlet distributions by using graph structure information. The key idea of the GKDE is to estimate prior Dirichlet distribution parameters for each node based on the class labels of training nodes (see Figure 3). Then, we use the estimated prior Dirichlet distribution in the training process to learn the following patterns: (i) nodes with a high vacuity will be shown far from training nodes; and (ii) nodes with a high dissonance will be shown near the boundaries of classes.

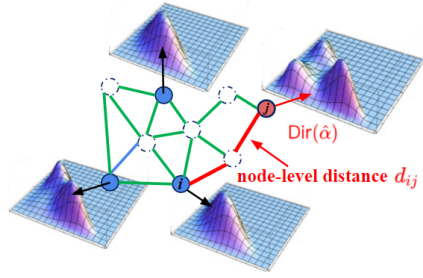

Figure 3: Illustration of GKDE. Estimate prior Dirichlet distribution $\text{Dir}(\hat{\alpha})$ for node $j$ (red) based on training nodes (blue) and graph structure information.

Based on SL, let each training node represent one evidence for its class label. Denote the contribution of evidence estimation for node $j$ from training node $i$ by $\mathbf{h}(y_i, d_{ij}) = [h_1, \dots, h_k, \dots, h_K] \in [0, 1]^K$, where $h_k(y_i, d_{ij})$ is obtained by:

$$h_k(y_i, d_{ij}) = \begin{cases} 0 & y_i \neq k \\ g(d_{ij}) & y_i = k, \end{cases} \quad (10)$$

$g(d_{ij}) = \frac{1}{\sigma\sqrt{2\pi}} \exp(-\frac{d_{ij}^2}{2\sigma^2})$ is the Gaussian kernel function used to estimate the distribution effect between nodes $i$ and $j$, and $d_{ij}$ means the **node-level distance** (**a shortest path between nodes $i$ and $j$**), and $\sigma$ is the bandwidth parameter. The prior evidence is estimated based GKDE: $\hat{e}_j = \sum_{i \in \mathbb{L}} \mathbf{h}(y_i, d_{ij})$, where $\mathbb{L}$ is a set of training nodes and the prior Dirichlet distribution $\hat{\boldsymbol{\alpha}}_j = \hat{e}_j + \mathbf{1}$. During the training process, we minimize the KL-divergence between model predictions of Dirichlet distribution and prior distribution: $\min \text{KL}[\text{Dir}(\boldsymbol{\alpha})\|\text{Dir}(\hat{\boldsymbol{\alpha}})]$. This process can prioritize the extent of data relevance based on the estimated evidential uncertainty, which is proven effective based on the proposition below.

**Proposition 1** *Given $L$ training nodes, for any testing nodes $i$ and $j$, let $\boldsymbol{d}_i = [d_{i1}, \dots, d_{iL}]$ be the vector of graph distances from nodes $i$ to training nodes and $\boldsymbol{d}_j = [d_{j1}, \dots, d_{jL}]$ be the graph distances from nodes $j$ to training nodes, where $d_{il}$ is the node-level distance between nodes $i$ and $l$. If for all $l \in \{1, \dots, L\}$, $d_{il} \geq d_{jl}$, then we have*

$$\hat{u}_{v_i} \geq \hat{u}_{v_j},$$

*where $\hat{u}_{v_i}$ and $\hat{u}_{v_j}$ refer to vacuity uncertainties of nodes $i$ and $j$ estimated based on GKDE.*

The proof for this proposition can be found in Appendix A.2. The above proposition shows that if a testing node is too far from training nodes, the vacuity will increase, implying that an OOD node is expected to have a high vacuity.

In addition, we designed a simple iterative knowledge distillation method [7] (i.e., Teacher Network) to refine the node-level classification probabilities. The key idea is to train our proposed model

(Student) to imitate the outputs of a pre-train a vanilla GNN (Teacher) by adding a regularization term of KL-divergence. This leads to solving the following optimization problem:

$$\min_{\boldsymbol{\theta}} \mathcal{L}(\boldsymbol{\theta}) + \lambda_1 \text{KL}[\text{Dir}(\boldsymbol{\alpha}) \| \text{Dir}(\hat{\boldsymbol{\alpha}})] + \lambda_2 \text{KL}[P(\mathbf{y} \mid A, \mathbf{r}; \mathcal{G}) \| P(\mathbf{y}|\hat{\mathbf{p}})], \tag{11}$$

where $\hat{\mathbf{p}}$ is the vanilla GNN's (Teacher) output and $\lambda_1$ and $\lambda_2$ are trade-off parameters.

# 6 Experiments

In this section, we conduct experiments on the tasks of misclassification and OOD detections to answer the following questions for semi-supervised node classification:

**Q1. Misclassification Detection:** What type of uncertainty is the most promising indicator of high confidence in node classification predictions?

**Q2. OOD Detection:** What type of uncertainty is a key indicator of accurate detection of OOD nodes?

**Q3. GKDE with Uncertainty Estimates:** How can GKDE help enhance prediction tasks with what types of uncertainty estimates?

Through extensive experiments, we found the following answers for the above questions:

**A1.** Dissonance (i.e., uncertainty due to conflicting evidence) is more effective than other uncertainty estimates in misclassification detection.

**A2.** Vacuity (i.e., uncertainty due to lack of confidence) is more effective than other uncertainty estimates in OOD detection.

**A3.** GKDE can indeed help improve the estimation quality of node-level Dirichlet distributions, resulting in a higher OOD detection.

## 6.1 Experiment Setup

**Datasets**: We used six datasets, including three citation network datasets [17] (i.e., Cora, Citeseer, Pubmed) and three new datasets [20] (i.e., Coauthor Physics, Amazon Computer, and Amazon Photo). We summarized the description and experimental setup of the used datasets in Appendix B.2[1].

**Comparing Schemes**: We conducted the extensive comparative performance analysis based on our proposed models and several state-of-the-art competitive counterparts. We implemented all models based on the most popular GNN model, GCN [12]. We compared our model (S-BGCN-T-K) against: (1) Softmax-based GCN [12] with uncertainty measured based on entropy; and (2) Drop-GCN that adapts the Monte-Carlo Dropout [5, 16] into the GCN model to learn probabilistic uncertainty; (3) EDL-GCN that adapts the EDL model [18] with GCN to estimate evidential uncertainty; (4) DPN-GCN that adapts the DPN [14] method with GCN to estimate probabilistic uncertainty. We evaluated the performance of all models considered using the area under the ROC (AUROC) curve and area under the Precision-Recall (AUPR) curve in both experiments [6].

## 6.2 Results

**Misclassification Detection.** The misclassification detection experiment involves detecting whether a given prediction is incorrect using an uncertainty estimate. Table 1 shows that S-BGCN-T-K outperforms all baseline models under the AUROC and AUPR for misclassification detection. The outperformance of dissonance-based detection is fairly impressive. This confirms that low dissonance (a small amount of conflicting evidence) is the key to maximize the accuracy of node classification prediction. We observe the following performance order: `Dissonance` > `Entropy` ≈ `Aleatoric` > `Vacuity` ≈ `Epistemic`, which is aligned with our conjecture: higher dissonance with conflicting prediction leads to higher misclassification detection. We also conducted experiments on additional three datasets and observed similar trends of the results, as demonstrated in Appendix C.

**OOD Detection.** This experiment involves detecting whether an input example is out-of-distribution (OOD) given an estimate of uncertainty. For semi-supervised node classification, we randomly selected one to four categories as OOD categories and trained the models based on training nodes of the other categories. Due to the space constraint, the experimental setup for the OOD detection is detailed in Appendix B.3.

In Table 2, across six network datasets, our vacuity-based detection significantly outperformed the other competitive methods, exceeding the performance of the epistemic uncertainty and other type of

Table 1: AUROC and AUPR for the Misclassification Detection.

| Data | Model | AUROC | | | | | AUPR | | | | | Acc |
|---|---|---|---|---|---|---|---|---|---|---|---|---|
| | | Va. | Dis. | Al. | Ep. | En. | Va. | Dis. | Al. | Ep. | En. | |
| Cora | S-BGCN-T-K | 70.6 | **82.4** | 75.3 | 68.8 | 77.7 | 90.3 | **95.4** | 92.4 | 87.8 | 93.4 | **82.0** |
| | EDL-GCN | 70.2 | 81.5 | - | - | 76.9 | 90.0 | 94.6 | - | - | 93.6 | 81.5 |
| | DPN-GCN | - | - | 78.3 | 75.5 | 77.3 | - | - | 92.4 | 92.0 | 92.4 | 80.8 |
| | Drop-GCN | - | - | 73.9 | 66.7 | 76.9 | - | - | 92.7 | 90.0 | 93.6 | 81.3 |
| | GCN | - | - | - | - | 79.6 | - | - | - | - | 94.1 | 81.5 |
| Citeseer | S-BGCN-T-K | 65.4 | **74.0** | 67.2 | 60.7 | 70.0 | 79.8 | **85.6** | 82.2 | 75.2 | 83.5 | **71.0** |
| | EDL-GCN | 64.9 | 73.6 | - | - | 69.6 | 79.2 | 84.6 | - | - | 82.9 | 70.2 |
| | DPN-GCN | - | - | 66.0 | 64.9 | 65.5 | - | - | 78.7 | 77.6 | 78.1 | 68.1 |
| | Drop-GCN | - | - | 66.4 | 60.8 | 69.8 | - | - | 82.3 | 77.8 | 83.7 | 70.9 |
| | GCN | - | - | - | - | 71.4 | - | - | - | - | 83.2 | 70.3 |
| Pubmed | S-BGCN-T-K | 64.1 | **73.3** | 69.3 | 64.2 | 70.7 | 85.6 | **90.8** | 88.8 | 86.1 | 89.2 | **79.3** |
| | EDL-GCN | 62.6 | 69.0 | - | - | 67.2 | 84.6 | 88.9 | - | - | 81.7 | 79.0 |
| | DPN-GCN | - | - | 72.7 | 69.2 | 72.5 | - | - | 87.8 | 86.8 | 87.7 | 77.1 |
| | Drop-GCN | - | - | 67.3 | 66.1 | 67.2 | - | - | 88.6 | 85.6 | 89.0 | 79.0 |
| | GCN | - | - | - | - | 68.5 | - | - | - | - | 89.2 | 79.0 |

Va.: Vacuity, Dis.: Dissonance, Al.: Aleatoric, Ep.: Epistemic, En.: Entropy

Table 2: AUROC and AUPR for the OOD Detection.

| Data | Model | AUROC | | | | | AUPR | | | | |
|---|---|---|---|---|---|---|---|---|---|---|---|
| | | Va. | Dis. | Al. | Ep. | En. | Va. | Dis. | Al. | Ep. | En. |
| Cora | S-BGCN-T-K | **87.6** | 75.5 | 85.5 | 70.8 | 84.8 | **78.4** | 49.0 | 75.3 | 44.5 | 73.1 |
| | EDL-GCN | 84.5 | 81.0 | | - | 83.3 | 74.2 | 53.2 | - | - | 71.4 |
| | DPN-GCN | - | - | 77.3 | 78.9 | 78.3 | - | - | 58.5 | 62.8 | 63.0 |
| | Drop-GCN | - | - | 81.9 | 70.5 | 80.9 | - | - | 69.7 | 44.2 | 67.2 |
| | GCN | - | - | - | - | 80.7 | - | - | - | - | 66.9 |
| Citeseer | S-BGCN-T-K | **84.8** | 55.2 | 78.4 | 55.1 | 74.0 | **86.8** | 54.1 | 80.8 | 55.8 | 74.0 |
| | EDL-GCN | 78.4 | 59.4 | - | - | 69.1 | 79.8 | 57.3 | - | - | 69.0 |
| | DPN-GCN | - | - | 68.3 | 72.2 | 69.5 | - | - | 68.5 | 72.1 | 70.3 |
| | Drop-GCN | - | - | 72.3 | 61.4 | 70.6 | - | - | 73.5 | 60.8 | 70.0 |
| | GCN | - | - | - | - | 70.8 | - | - | - | - | 70.2 |
| Pubmed | S-BGCN-T-K | **74.6** | 67.9 | 71.8 | 59.2 | 72.2 | **69.6** | 52.9 | 63.6 | 44.0 | 56.5 |
| | EDL-GCN | 71.5 | 68.2 | - | - | 70.5 | 65.3 | 53.1 | - | - | 55.0 |
| | DPN-GCN | - | - | 63.5 | 63.7 | 63.5 | - | - | 50.7 | 53.9 | 51.1 |
| | Drop-GCN | - | - | 68.7 | 60.8 | 66.7 | - | - | 59.7 | 46.7 | 54.8 |
| | GCN | - | - | - | - | 68.3 | - | - | - | - | 55.3 |
| Amazon Photo | S-BGCN-T-K | **93.4** | 76.4 | 91.4 | 32.2 | 91.4 | **94.8** | 68.0 | 92.3 | 42.3 | 92.5 |
| | EDL-GCN | 63.4 | 78.1 | - | - | 79.2 | 66.2 | 74.8 | - | - | 81.2 |
| | DPN-GCN | - | - | 83.6 | 83.6 | 83.6 | - | - | 82.6 | 82.4 | 82.5 |
| | Drop-GCN | - | - | 84.5 | 58.7 | 84.3 | - | - | 87.0 | 57.7 | 86.9 |
| | GCN | - | - | - | - | 84.4 | - | - | - | - | 87.0 |
| Amazon Computer | S-BGCN-T-K | **82.3** | 76.6 | 80.9 | 55.4 | 80.9 | **70.5** | 52.8 | 60.9 | 35.9 | 60.6 |
| | EDL-GCN | 53.2 | 70.1 | - | - | 70.0 | 33.2 | 43.9 | - | - | 45.7 |
| | DPN-GCN | - | - | 77.6 | 77.7 | 77.7 | - | - | 50.8 | 51.2 | 51.0 |
| | Drop-GCN | - | - | 74.4 | 70.5 | 74.3 | - | - | 50.0 | 46.7 | 49.8 |
| | GCN | - | - | - | - | 74.0 | - | - | - | - | 48.7 |
| Coauthor Physics | S-BGCN-T-K | **91.3** | 87.6 | 89.7 | 61.8 | 89.8 | **72.2** | 56.6 | 68.1 | 25.9 | 67.9 |
| | EDL-GCN | 88.2 | 85.8 | - | - | 87.6 | 67.1 | 51.2 | - | - | 62.1 |
| | DPN-GCN | - | - | 85.5 | 85.6 | 85.5 | - | - | 59.8 | 60.2 | 59.8 |
| | Drop-GCN | - | - | 89.2 | 78.4 | 89.3 | - | - | 66.6 | 37.1 | 66.5 |
| | GCN | - | - | - | - | 89.1 | - | - | - | - | 64.0 |

Va.: Vacuity, Dis.: Dissonance, Al.: Aleatoric, Ep.: Epistemic, En.: Entropy

uncertainties. This demonstrates that vacuity-based model is more effective than other uncertainty estimates-based counterparts in increasing OOD detection. We observed the following performance order: `Vacuity > Entropy ≈ Aleatoric > Epistemic ≈ Dissonance`, which is consistent with the theoretical results as shown in Theorem 1.

**Ablation Study.** We conducted additional experiments (see Table 3) in order to demonstrate the contributions of the key technical components, including GKDE, Teacher Network, and subjective Bayesian framework. The key findings obtained from this experiment are: (1) GKDE can enhance the OOD detection (i.e., 30% increase with vacuity), which is consistent with our theoretical proof about the outperformance of GKDE in uncertainty estimation, i.e., OOD nodes have a higher vacuity than other nodes; and (2) the Teacher Network can further improve the node classification accuracy.

## 6.3 Why is Epistemic Uncertainty Less Effective than Vacuity?

Although epistemic uncertainty is known to be effective to improve OOD detection [5, 11] in computer vision applications, our results demonstrate it is less effective than our vacuity-based approach. The first potential reason is that epistemic uncertainty is always smaller than vacuity (From Theorem 1), which potentially indicates that epistemic may capture less information related to OOD. Another potential reason is that the previous success of epistemic uncertainty for OOD detection is limited to supervised learning in computer vision applications, but its effectiveness for OOD detection was not

sufficiently validated in semi-supervised learning tasks. Recall that epistemic uncertainty (i.e., model uncertainty) is calculated based on mutual information (see Eq. (6)). In a semi-supervised setting, the features of unlabeled nodes are also fed to a model for training process to provide the model with a high confidence on its output. For example, the model output $P(\mathbf{y}|A, \mathbf{r}; \theta)$ would not change too much even with differently sampled parameters $\boldsymbol{\theta}$, i.e., $P(\mathbf{y}|A, \mathbf{r}; \theta^{(i)}) \approx P(\mathbf{y}|A, \mathbf{r}; \theta^{(j)})$, which result in a low epistemic uncertainty. We also designed a semi-supervised learning experiment for image classification and observed a consistent pattern with the results demonstrated in Appendix C.6.

Table 3: Ablation study of our proposed models: (1) `S-GCN`: Subjective GCN with vacuity and dissonance estimation; (2) `S-BGCN`: S-GCN with Bayesian framework; (3) `S-BGCN-T`: S-BGCN with a Teacher Network; (4) `S-BGCN-T-K`: S-BGCN-T with GKDE to improve uncertainty estimation.

| Data | Model | AUROC (Misclassification Detection) | | | | | AUPR (Misclassification Detection) | | | | | Acc |
|---|---|---|---|---|---|---|---|---|---|---|---|---|
| | | Va. | Dis. | Al. | Ep. | En. | Va. | Dis. | Al. | Ep. | En. | |
| Cora | S-BGCN-T-K | 70.6 | 82.4 | 75.3 | 68.8 | 77.7 | 90.3 | **95.4** | 92.4 | 87.8 | 93.4 | 82.0 |
| | S-BGCN-T | 70.8 | **82.5** | 75.3 | 68.9 | 77.8 | 90.4 | **95.4** | 92.6 | 88.0 | 93.4 | **82.2** |
| | S-BGCN | 69.8 | 81.4 | 73.9 | 66.7 | 76.9 | 89.4 | 94.3 | 92.3 | 88.0 | 93.1 | 81.2 |
| | S-GCN | 70.2 | 81.5 | - | - | 76.9 | 90.0 | 94.6 | - | - | 93.6 | 81.5 |
| | | AUROC (OOD Detection) | | | | | AUPR (OOD Detection) | | | | | |
| Amazon Photo | S-BGCN-T-K | **93.4** | 76.4 | 91.4 | 32.2 | 91.4 | **94.8** | 68.0 | 92.3 | 42.3 | 92.5 | - |
| | S-BGCN-T | 64.0 | 77.5 | 79.9 | 52.6 | 79.8 | 67.0 | 75.3 | 82.0 | 53.7 | 81.9 | - |
| | S-BGCN | 63.0 | 76.6 | 79.8 | 52.7 | 79.7 | 66.5 | 75.1 | 82.1 | 53.9 | 81.7 | - |
| | S-GCN | 64.0 | 77.1 | - | - | 79.6 | 67.0 | 74.9 | - | - | 81.6 | - |

Va.: Vacuity, Dis.: Dissonance, Al.: Aleatoric, Ep.: Epistemic, En.: Entropy

# 7   Conclusion

In this work, we proposed a multi-source uncertainty framework of GNNs for semi-supervised node classification. Our proposed framework provides an effective way of predicting node classification and out-of-distribution detection considering multiple types of uncertainty. We leveraged various types of uncertainty estimates from both DL and evidence/belief theory domains. Through our extensive experiments, we found that dissonance-based detection yielded the best performance on misclassification detection while vacuity-based detection performed the best for OOD detection, compared to other competitive counterparts. In particular, it was noticeable that applying GKDE and the Teacher network further enhanced the accuracy in node classification and uncertainty estimates.

## Acknowledgments

We would like to thank Yuzhe Ou for providing proof suggestions. This work is supported by the National Science Foundation (NSF) under Grant No #1815696 and #1750911.

## Broader Impact

In this paper, we propose a uncertainty-aware semi-supervised learning framework of GNN for predicting multi-dimensional uncertainties for the task of semi-supervised node classification. Our proposed framework can be applied to a wide range of applications, including computer vision, natural language processing, recommendation systems, traffic prediction, generative models and many more [23]. Our proposed framework can be applied to predict multiple uncertainties of different roots for GNNs in these applications, improving the understanding of individual decisions, as well as the underlying models. While there will be important impacts resulting from the use of GNNs in general, our focus in this work is on investigating the impact of using our method to predict multi-source uncertainties for such systems. The additional benefits of this method include improvement of safety and transparency in decision-critical applications to avoid overconfident prediction, which can easily lead to misclassification.

We see promising research opportunities that can adopt our uncertainty framework, such as investigating whether this uncertainty framework can further enhance misclassification detection or OOD detection. To mitigate the risk from different types of uncertainties, we encourage future research to understand the impacts of this proposed uncertainty framework to solve other real world problems.

## Footnotes

[1]The source code and datasets are accessible at https://github.com/zxj32/uncertainty-GNN

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
