[Supplementary Material]

# Supplementary Material

## A Proofs

### A.1 Theorem 1's Proof

**Theorem 1.** *We consider a simplified scenario, where a multinomial random variable $y$ follows a $K$-class categorical distribution: $y \sim Cal(\boldsymbol{p})$, the class probabilities $\boldsymbol{p}$ follow a Dirichlet distribution: $\boldsymbol{p} \sim Dir(\boldsymbol{\alpha})$, and $\boldsymbol{\alpha}$ refer to the Dirichlet parameters. Given a total Dirichlet strength $S = \sum_{i=1}^{K} \alpha_i$, for any opinion $\omega$ on a multinomial random variable $y$, we have*

1. *General relations on all prediction scenarios.*

    *(a) $u_v + u_{diss} \leq 1$; (b) $u_v > u_{epis}$.*

2. *Special relations on the OOD and the CP.*

    *(a) For an OOD sample with a uniform prediction (i.e., $\alpha = [1, \ldots, 1]$), we have*
    $$1 = u_v = u_{en} > u_{alea} > u_{epis} > u_{diss} = 0$$

    *(b) For an in-distribution sample with a conflicting prediction (i.e., $\alpha = [\alpha_1, \ldots, \alpha_K]$ with $\alpha_1 = \alpha_2 = \cdots = \alpha_K$, if $S \to \infty$), we have*
    $$u_{en} = 1, \lim_{S \to \infty} u_{diss} = \lim_{S \to \infty} u_{alea} = 1, \lim_{S \to \infty} u_v = \lim_{S \to \infty} u_{epis} = 0$$
    *with $u_{en} > u_{alea} > u_{diss} > u_v > u_{epis}$.*

**Interpretation**. **Theorem 1.1 (a)** implies that increases in both uncertainty types may not happen at the same time. A higher vacuity leads to a lower dissonance, and vice versa (a higher dissonance leads to a lower vacuity). This indicates that a high dissonance only occurs only when a large amount of evidence is available and the vacuity is low. **Theorem 1.1 (b)** shows relationships between vacuity and epistemic uncertainty in which vacuity is an upper bound of epistemic uncertainty. Although some existing approaches [5, 17] treat epistemic uncertainty the same as vacuity, it is not necessarily true except for an extreme case where a sufficiently large amount of evidence available, making vacuity close to zero. **Theorem 1.2 (a) and (b)** explain how entropy differs from vacuity and/or dissonance. We observe that entropy is 1 when either vacuity or dissonance is 0. This implies that entropy cannot distinguish different types of uncertainty due to different root causes. For example, a high entropy is observed when an example is an either OOD or misclassified example. Similarly, a high aleatoric uncertainty value and a low epistemic uncertainty value are observed under both cases. However, vacuity and dissonance can capture different causes of uncertainty due to lack of information and knowledge and to conflicting evidence, respectively. For example, an OOD objects typically show a high vacuity value and a low dissonance value while a conflicting prediction exhibits a low vacuity and a high dissonance.

*Proof.* 1. (a) Let the opinion $\omega = [b_1, \ldots, b_K, u_v]$, where $K$ is the number of classes, $b_i$ is the belief for class $i$, $u_v$ is the uncertainty mass (vacuity), and $\sum_{i=1}^{K} b_i + u_v = 1$. Dissonance has a upper bound with

$$
\begin{aligned}
u_{diss} &= \sum_{i=1}^{K} \Big( \frac{b_i \sum_{j=1, j \neq i}^{K} b_j \mathrm{Bal}(b_i, b_j)}{\sum_{j=1, j \neq i}^{K} b_j} \Big) \qquad (1) \\
&\leq \sum_{i=1}^{K} \Big( \frac{b_i \sum_{j=1, j \neq i}^{K} b_j}{\sum_{j=1, j \neq i}^{K} b_j} \Big), \quad (\text{since } 0 \leq \mathrm{Bal}(b_i, b_j) \leq 1) \\
&= \sum_{i=1}^{K} b_i,
\end{aligned}
$$

where $\text{Bal}(b_i, b_j)$ is the relative mass balance, then we have

$$u_v + u_{diss} \leq \sum_{i=1}^{K} b_i + u_v = 1. \tag{2}$$

1. (b) For the multinomial random variable $y$, we have

$$y \sim \text{Cal}(\mathbf{p}), \quad \mathbf{p} \sim \text{Dir}(\boldsymbol{\alpha}), \tag{3}$$

where $\text{Cal}(\mathbf{p})$ is the categorical distribution and $\text{Dir}(\boldsymbol{\alpha})$ is Dirichlet distribution. Then we have

$$\text{Prob}(y|\boldsymbol{\alpha}) = \int \text{Prob}(y|\mathbf{p})\text{Prob}(\mathbf{p}|\boldsymbol{\alpha})d\mathbf{p}, \tag{4}$$

and the epistemic uncertainty is estimated by mutual information,

$$\mathcal{I}[y, \mathbf{p}|\boldsymbol{\alpha}] = \mathcal{H}\Big[\mathbb{E}_{\text{Prob}(\mathbf{p}|\boldsymbol{\alpha})}[P(y|\mathbf{p})]\Big] - \mathbb{E}_{\text{Prob}(\mathbf{p}|\boldsymbol{\alpha})}\Big[\mathcal{H}[P(y|\mathbf{p})]\Big]. \tag{5}$$

Now we consider another measure of ensemble diversity: *Expected Pairwise KL-Divergence* between each model in the ensemble. Here the expected pairwise KL-Divergence between two independent distributions, including $P(y|\mathbf{p}_1)$ and $P(y|\mathbf{p}_2)$, where $\mathbf{p}_1$ and $\mathbf{p}_2$ are two independent samples from $\text{Prob}(\mathbf{p}|\boldsymbol{\alpha})$, can be computed,

$$
\begin{aligned}
\mathcal{K}[y, \mathbf{p}|\boldsymbol{\alpha}] &= \mathbb{E}_{\text{Prob}(\mathbf{p}_1|\boldsymbol{\alpha}\text{Prob}(\mathbf{p}_2|\boldsymbol{\alpha})}\Big[KL[P(y|\mathbf{p}_1)\|P(y|\mathbf{p}_2)]\Big] \\
&= -\sum_{i=1}^{K}\mathbb{E}_{\text{Prob}(\mathbf{p}_1|\boldsymbol{\alpha})}[P(y|\mathbf{p}_1)]\mathbb{E}_{\text{Prob}(\mathbf{p}_2|\boldsymbol{\alpha})}[\ln P(y|\mathbf{p}_2)] - \mathbb{E}_{\text{Prob}(\mathbf{p}|\boldsymbol{\alpha})}\Big[\mathcal{H}[P(y|\mathbf{p})]\Big] \\
&\geq \mathcal{I}[y, \mathbf{p}|\boldsymbol{\alpha}],
\end{aligned}
\tag{6}
$$

where $\mathcal{I}[y, \mathbf{p}_1|\boldsymbol{\alpha}] = \mathcal{I}[y, \mathbf{p}_2|\boldsymbol{\alpha}]$. We consider Dirichlet ensemble, the *Expected Pairwise KL Divergence*,

$$
\begin{aligned}
\mathcal{K}[y, \mathbf{p}|\boldsymbol{\alpha}] &= -\sum_{i=1}^{K}\frac{\alpha_i}{S}\Big(\psi(\alpha_i) - \psi(S)\Big) - \sum_{i=1}^{K}-\frac{\alpha_i}{S}\Big(\psi(\alpha_i + 1) - \psi(S + 1)\Big) \\
&= \frac{K-1}{S},
\end{aligned}
\tag{7}
$$

where $S = \sum_{i=1}^{K}\alpha_i$ and $\psi(\cdot)$ is the *digamma Function*, which is the derivative of the natural logarithm of the gamma function. Now we obtain the relations between vacuity and epistemic,

$$\underbrace{\frac{K}{S}}_{\text{Vacuity}} > \mathcal{K}[y, \mathbf{p}|\boldsymbol{\alpha}] = \frac{K-1}{S} \geq \underbrace{\mathcal{I}[y, \mathbf{p}|\boldsymbol{\alpha}]}_{\text{Epistemic}}. \tag{8}$$

2. (a) For an out-of-distribution sample, $\alpha = [1, \ldots, 1]$, the vacuity can be calculated as

$$u_v = \frac{K}{\sum_{i=1}^{K}\alpha_i} = \frac{K}{K} = 1, \tag{9}$$

and the belief mass $b_i = (\alpha_i - 1)/\sum_{i=1}^{K}\alpha_i = 0$, we estimate dissonance,

$$u_{diss} = \sum_{i=1}^{K}\Big(\frac{b_i \sum_{j=1,j\neq i}^{K} b_j \text{Bal}(b_i, b_j)}{\sum_{j=1,j\neq i}^{K} b_j}\Big) = 0. \tag{10}$$

Given the expected probability $\hat{p} = [1/K, \ldots, 1/K]^{\top}$, the entropy is calculated based on $\log_K$,

$$u_{en} = \mathcal{H}[\hat{p}] = -\sum_{i=1}^{K}\hat{p}_i \log_K \hat{p}_i = -\sum_{i=1}^{K}\frac{1}{K}\log_K\frac{1}{K} = \log_K\frac{1}{K}^{-1} = \log_K K = 1, \tag{11}$$

where $\mathcal{H}(\cdot)$ is the entropy. Based on Dirichlet distribution, the aleatoric uncertainty refers to the expected entropy,

$$
\begin{aligned}
u_{alea} &= \mathbb{E}_{p\sim\text{Dir}(\alpha)}[\mathcal{H}[p]] \qquad\qquad\qquad\qquad\qquad\qquad\qquad\qquad\qquad (12)\\
&= -\sum_{i=1}^{K} \frac{\Gamma(S)}{\prod_{i=1}^{K}\Gamma(\alpha_i)} \int_{S_K} p_i \log_K p_i \prod_{i=1}^{K} p_i^{\alpha_i-1} d\boldsymbol{p}\\
&= -\frac{1}{\ln K} \sum_{i=1}^{K} \frac{\Gamma(S)}{\prod_{i=1}^{K}\Gamma(\alpha_i)} \int_{S_K} p_i \ln p_i \prod_{i=1}^{K} p_i^{\alpha_i-1} d\boldsymbol{p}\\
&= -\frac{1}{\ln K} \sum_{i=1}^{K} \frac{\alpha_i}{S} \frac{\Gamma(S+1)}{\Gamma(\alpha_i+1)\prod_{i'=1,\neq i}^{K}\Gamma(\alpha_{i'})} \int_{S_K} p_i^{\alpha_i} \ln p_i \prod_{i'=1,\neq i}^{K} p_{i'}^{\alpha_{i'}-1} d\boldsymbol{p}\\
&= \frac{1}{\ln K} \sum_{i=1}^{K} \frac{\alpha_i}{S} \big(\psi(S+1) - \psi(\alpha_i+1)\big)\\
&= \frac{1}{\ln K} \sum_{i=1}^{K} \frac{1}{K}(\psi(K+1) - \psi(2))\\
&= \frac{1}{\ln K}(\psi(K+1) - \psi(2))\\
&= \frac{1}{\ln K}\Big(\psi(2) + \sum_{k=2}^{K} \frac{1}{k} - \psi(2)\Big)\\
&= \frac{1}{\ln K} \sum_{k=2}^{K} \frac{1}{k} < \frac{1}{\ln K} \ln K = 1,
\end{aligned}
$$

where $S = \sum_{i=1}^{K} \alpha_i$, $\boldsymbol{p} = [p_1, \ldots, p_K]^\top$, and $K \geq 2$ is the number of category. The epistemic uncertainty can be calculated via the mutual information,

$$
\begin{aligned}
u_{epis} &= \mathcal{H}[\mathbb{E}_{p\sim\text{Dir}(\alpha)}[p]] - \mathbb{E}_{p\sim\text{Dir}(\alpha)}[\mathcal{H}[p]] \qquad\qquad\qquad\qquad (13)\\
&= \mathcal{H}[\hat{p}] - u_{alea}\\
&= 1 - \frac{1}{\ln K} \sum_{k=2}^{K} \frac{1}{k} < 1.
\end{aligned}
$$

To compare aleatoric uncertainty with epistemic uncertainty, we first prove that aleatoric uncertainty (Eq. (13)) is monotonically increasing and converging to 1 as $K$ increases. Based on *Lemma 1*, we have

$$
\Big(\ln(K+1) - \ln K\Big) \sum_{k=2}^{K} \frac{1}{k} < \frac{\ln K}{K+1}
$$

$$
\Rightarrow \ln(K+1) \sum_{k=2}^{K} \frac{1}{k} < \ln K \Big(\sum_{k=2}^{K} \frac{1}{k} + \frac{1}{K+1}\Big) = \ln K \sum_{k=2}^{K+1} \frac{1}{k}
$$

$$
\Rightarrow \frac{1}{\ln K} \sum_{k=2}^{K} \frac{1}{k} < \frac{1}{\ln(K+1)} \sum_{k=2}^{K+1} \frac{1}{k}. \qquad\qquad\qquad\qquad\qquad (14)
$$

Based on Eq. (14) and Eq. (13), we prove that aleatoric uncertainty is monotonically increasing with respect to $K$. So the minimum aleatoric can be shown to be $\frac{1}{\ln 2} \frac{1}{2}$, when $K = 2$.

Similarly, for epistemic uncertainty, which is monotonically decreasing as $K$ increases based on *Lemma 1*, the maximum epistemic can be shown to be $1 - \frac{1}{\ln 2} \frac{1}{2}$ when $K = 2$. Then we have,

$$
u_{alea} \geq \frac{1}{\ln 2} \frac{1}{2} > 1 - \frac{1}{2\ln 2} \geq u_{epis} \qquad\qquad\qquad\qquad\qquad (15)
$$

Therefore, we prove that $1 = u_v = u_{en} > u_{alea} > u_{epis} > u_{diss} = 0$.

2. (b) For a conflicting prediction, i.e., $\alpha = [\alpha_1, \ldots, \alpha_K]$, with $\alpha_1 = \alpha_2 = \cdots = \alpha_K = C$, and $S = \sum_{i=1}^{K} \alpha_i = CK$, the expected probability $\hat{p} = [1/K, \ldots, 1/K]^\top$, the belief mass $b_i = (\alpha_i - 1)/S$, and the vacuity can be calculated as

$$u_v \quad = \quad \frac{K}{S} \xrightarrow{S \to \infty} 0, \tag{16}$$

and the dissonance can be calculated as

$$
\begin{aligned}
u_{diss} \quad &= \quad \sum_{i=1}^{K} \left( \frac{b_i \sum_{j=1, j \neq i}^{K} b_j \mathrm{Bal}(b_i, b_j)}{\sum_{j=1, j \neq i}^{K} b_j} \right) = \sum_{i=1}^{K} b_i \\
&= \quad \sum_{i=1}^{K} \left( \frac{a_i - 1}{\sum_{i=1}^{K} a_i} \right) \\
&= \quad \frac{\sum_{i=1}^{K} a_i - k}{\sum_{i=1}^{K} a_i} \\
&= \quad 1 - \frac{K}{S} \xrightarrow{S \to \infty} 1.
\end{aligned}
\tag{17}
$$

Given the expected probability $\hat{p} = [1/K, \ldots, 1/K]^\top$, the entropy can be calculated based on Dirichlet distribution,

$$u_{en} \quad = \quad \mathcal{H}[\hat{p}] = \sum_{i=1}^{K} \hat{p}_i \log_K \hat{p}_i = 1, \tag{18}$$

and the aleatoric uncertainty is estimated as the expected entropy,

$$
\begin{aligned}
u_{alea} \quad &= \quad \mathbb{E}_{p \sim \mathrm{Dir}(\alpha)}[\mathcal{H}[p]] \\
&= \quad -\sum_{i=1}^{K} \frac{\Gamma(S)}{\prod_{i=1}^{K} \Gamma(\alpha_i)} \int_{S_K} p_i \log_K p_i \prod_{i=1}^{K} p_i^{\alpha_i - 1} d\boldsymbol{p} \\
&= \quad -\frac{1}{\ln K} \sum_{i=1}^{K} \frac{\Gamma(S)}{\prod_{i=1}^{K} \Gamma(\alpha_i)} \int_{S_K} p_i \ln p_i \prod_{i=1}^{K} p_i^{\alpha_i - 1} d\boldsymbol{p} \\
&= \quad -\frac{1}{\ln K} \sum_{i=1}^{K} \frac{\alpha_i}{S} \frac{\Gamma(S+1)}{\Gamma(\alpha_i + 1) \prod_{i'=1, \neq i}^{K} \Gamma(\alpha_{i'})} \int_{S_K} p_i^{\alpha_i} \ln p_i \prod_{i'=1, \neq i}^{K} p_{i'}^{\alpha_{i'} - 1} d\boldsymbol{p} \\
&= \quad \frac{1}{\ln K} \sum_{i=1}^{K} \frac{\alpha_i}{S} \left( \psi(S+1) - \psi(\alpha_i + 1) \right) \\
&= \quad \frac{1}{\ln K} \sum_{i=1}^{K} \frac{1}{K} \left( \psi(S+1) - \psi(C+1) \right) \\
&= \quad \frac{1}{\ln K} \left( \psi(S+1) - \psi(C+1) \right) \\
&= \quad \frac{1}{\ln K} \left( \psi(C+1) + \sum_{k=C+1}^{S} \frac{1}{k} - \psi(C+1) \right) \\
&= \quad \frac{1}{\ln K} \sum_{k=C+1}^{S} \frac{1}{k} \xrightarrow{S \to \infty} 1.
\end{aligned}
\tag{19}
$$

The epistemic uncertainty can be calculated via mutual information,

$$
\begin{aligned}
u_{epis} \quad &= \quad \mathcal{H}[\mathbb{E}_{p \sim \mathrm{Dir}(\alpha)}[p]] - \mathbb{E}_{p \sim \mathrm{Dir}(\alpha)}[\mathcal{H}[p]] \\
&= \quad \mathcal{H}[\hat{p}] - u_{alea} \\
&= \quad 1 - \frac{1}{\ln K} \sum_{k=C+1}^{S} \frac{1}{k} \xrightarrow{S \to \infty} 0.
\end{aligned}
\tag{20}
$$

Now we compare aleatoric uncertainty with vacuity,

$$
\begin{aligned}
u_{alea} &= \frac{1}{\ln K} \sum_{k=C+1}^{S} \frac{1}{k} \\
&= \frac{1}{\ln K} \sum_{k=C+1}^{CK} \frac{1}{k} \\
&= \frac{\ln(CK+1) - \ln(C+1)}{\ln K} \\
&= \frac{\ln(K - \frac{K-1}{C+1})}{\ln K} \\
&> \frac{\ln(K - \frac{K-1}{2})}{\ln K} \\
&= \frac{\ln(4/K + 4/K + 1/2)}{\ln K} \\
&\geq \frac{\ln[3(4/K + 4/K + 1/2)^{\frac{1}{3}}]}{\ln K} \\
&= \frac{\ln 3 + \frac{1}{3}\ln(\frac{K^2}{32})}{\ln K} \\
&= \frac{\ln 3 + \frac{2}{3}\ln K - \frac{1}{3}\ln 32}{\ln K} > \frac{2}{3}.
\end{aligned}
\tag{21}
$$

Based on Eq. (22), when $C > \frac{3}{2}$, we have

$$
u_{alea} > \frac{2}{3} > \frac{1}{C} = u_v
\tag{22}
$$

We have already proved that $u_v > u_{epis}$, when $u_{en} = 1$, we have $u_{alea} > u_{diss}$ Therefore, we prove that $u_{en} > u_{alea} > u_{diss} > u_v > u_{epis}$ with $u_{en} = 1, u_{diss} \to 1, u_{alea} \to 1, u_v \to 0, u_{epis} \to 0$ $\qquad\square$

**Lemma 1.** *For all integer $N \geq 2$, we have $\sum_{n=2}^{N} \frac{1}{n} < \frac{\ln N}{(N+1)\ln(\frac{N+1}{N})}$.*

*Proof.* We will prove by induction that, for all integer $N \geq 2$,

$$
\sum_{n=2}^{N} \frac{1}{n} < \frac{\ln N}{(N+1)\ln(\frac{N+1}{N})}.
\tag{23}
$$

*Base case*: When $N = 2$, we have $\frac{1}{2} < \frac{\ln 2}{3\ln\frac{3}{2}}$ and Eq. (23) is true for $N = 2$.

*Induction step*: Let the integer $K \geq 2$ is given and suppose Eq. (23) is true for $N = K$, then

$$
\sum_{k=2}^{K+1} \frac{1}{k} = \frac{1}{K+1} + \sum_{k=2}^{K} \frac{1}{k} < \frac{1}{K+1} + \frac{\ln K}{(K+1)\ln(\frac{K+1}{K})} = \frac{\ln(K+1)}{(K+1)\ln(\frac{K+1}{K})}.
\tag{24}
$$

Denote that $g(x) = (x+1)\ln(\frac{x+1}{x})$ with $x > 2$. We get its derivative, $g'(x) = \ln(1 + \frac{1}{x}) - \frac{1}{x} < 0$, such that $g(x)$ is monotonically decreasing, which results in $g(K) > g(K+1)$. Based on Eq. (24) we have,

$$
\sum_{k=2}^{K+1} \frac{1}{k} < \frac{\ln(K+1)}{g(K)} < \frac{\ln(K+1)}{g(K+1)} = \frac{\ln(K+1)}{(K+2)\ln(\frac{K+2}{K+1})}.
\tag{25}
$$

Thus, Eq. (23) holds for $N = K + 1$, and the proof of the induction step is complete.

*Conclusion*: By the principle of induction, Eq. (23) is true for all integer $N \geq 2$. $\qquad\square$

## A.2 Proposition 1's Proof

**Proposition 1.** *Given $L$ training nodes, for any testing nodes $i$ and $j$, let $\boldsymbol{d}_i = [d_{i1}, \ldots, d_{iL}]$ is the graph distances from nodes $i$ to training nodes, and $\boldsymbol{d}_j = [d_{j1}, \ldots, d_{jL}]$ is the graph distances from nodes $j$ to training nodes, where $d_{il}$ is the node-level distance between nodes $i$ and $l$. If for all $l \in \{1, \ldots, L\}$, $d_{il} \geq d_{jl}$, then we have*

$$\hat{u}_{v_i} \geq \hat{u}_{v_j}$$

*where $\hat{u}_{v_i}$ and $\hat{u}_{v_j}$ are vacuity uncertainties estimated of nodes $i$ and $j$ based on GKDE.*

**Interpretation**. From the above proposition, if a testing node is too distant (far away) from training nodes, the vacuity increases, indicating that an OOD node is expected to have a high vacuity value.

*Proof.* Let $\boldsymbol{y} = [y_1, \ldots, y_L]$ be the label vector for training nodes. Based on GKDE, the evidence contribution for the node $i$ and a training node $l \in \{1, \ldots, L|\}$ is $\boldsymbol{h}(y_l, d_{il}) = [h_1(y_l, d_{il}), \ldots, h_K(y_l, d_{il})]$, where

$$h_k(y_l, d_{il}) = \begin{cases} 0 & y_l \neq k \\ g(d_{il}) = \frac{1}{\sigma\sqrt{2\pi}} \exp(-\frac{d_{il}^2}{2\sigma^2}) & y_l = k \end{cases}, \tag{26}$$

and the prior evidence can be estimated based GKDE:

$$\hat{e}_i = \sum_{m=1}^{L} \sum_{k=1}^{K} h_k(y_l, d_{il}), \tag{27}$$

where $\hat{e}_i = [e_{i1}, \ldots, e_{iK}]$. Since each training node only contributes the same evidence based on its label based on Eq. (26), the total evidence is estimated by all the contributing evidence as

$$\sum_{k=1}^{K} e_{ik} = \sum_{m=1}^{L} \frac{1}{\sigma\sqrt{2\pi}} \exp(-\frac{d_{il}^2}{2\sigma^2}), \quad \sum_{k=1}^{K} e_{jk} = \sum_{m=1}^{L} \frac{1}{\sigma\sqrt{2\pi}} \exp(-\frac{d_{jl}^2}{2\sigma^2}), \tag{28}$$

where the vacuity values for node $i$ and node $j$ based on GKDE are,

$$\hat{u}_{v_i} = \frac{K}{\sum_{k=1}^{K} e_{ik} + K}, \quad \hat{u}_{v_j} = \frac{K}{\sum_{k=1}^{K} e_{jk} + K}. \tag{29}$$

Now, we prove Eq. (29) above. If $d_{il} \geq d_{jl}$ for $\forall l \in \{1, \ldots, L\}$, we have

$$\begin{aligned} \sum_{k=1}^{K} e_{ik} &= \sum_{m=1}^{L} \frac{1}{\sigma\sqrt{2\pi}} \exp(-\frac{d_{il}^2}{2\sigma^2}) \\ &\leq \sum_{m=1}^{L} \frac{1}{\sigma\sqrt{2\pi}} \exp(-\frac{d_{jl}^2}{2\sigma^2}) \\ &= \sum_{k=1}^{K} e_{jk}, \end{aligned} \tag{30}$$

such that

$$\hat{u}_{v_i} = \frac{K}{\sum_{k=1}^{K} e_{ik} + K} \geq \frac{K}{\sum_{k=1}^{K} e_{jk} + K} = \hat{u}_{v_j}. \tag{31}$$

$\square$

# B  Additional Experimental Details

## B.1  Source code

The source code and datasets are accessible at https://github.com/zxj32/uncertainty-GNN

## B.2  Description of Datasets

Table 1: Description of datasets and their experimental setup for the node classification prediction.

|                    | Cora  | Citeseer | Pubmed | Co. Physics | Ama.Computer | Ama.Photo |
|--------------------|-------|----------|--------|-------------|--------------|-----------|
| #Nodes             | 2,708 | 3,327    | 19,717 | 34, 493     | 13, 381      | 7, 487    |
| #Edges             | 5,429 | 4,732    | 44,338 | 282, 455    | 259, 159     | 126, 530  |
| #Classes           | 7     | 6        | 3      | 5           | 10           | 8         |
| #Features          | 1,433 | 3,703    | 500    | 8,415       | 767          | 745       |
| #Training nodes    | 140   | 120      | 60     | 100         | 200          | 160       |
| #Validation nodes  | 500   | 500      | 500    | 500         | 500          | 500       |
| #Test nodes        | 1,000 | 1,000    | 1,000  | 1000        | 1,000        | 1000      |

**Cora, Citeseer, and Pubmed** [16]: These are citation network datasets, where each network is a directed network in which a node represents a document and an edge is a citation link, meaning that there exists an edge when $A$ document cites $B$ document, or vice-versa with a direction. Each node's feature vector contains a bag-of-words representation of a document. For simplicity, we don't discriminate the direction of links and treat citation links as undirected edges and construct a binary, symmetric adjacency matrix $\mathbf{A}$. Each node is labeled with the class to which it belongs.

**Coauthor Physics, Amazon Computers, and Amazon Photo** [18]: Coauthor Physics is the dataset for co-authorship graphs based on the Microsoft Academic Graph from the KDD Cup 2016 Challenge[1]. In the graphs, a node is an author and an edge exists when two authors co-author a paper. A node's features represent the keywords of its papers and the node's class label indicates its most active field of study. Amazon Computers and Amazon Photo are the segments of an Amazon co-purchase graph [13], where a node is a good (i.e., product), an edge exists when two goods are frequently bought together. A node's features are bag-of-words representation of product reviews and the node's class label is the product category.

For all the used datasets, we deal with undirected graphs with 20 training nodes for each category. We chose the same dataset splits as in [21] with an additional validation node set of 500 labeled examples for the hyperparameter obtained from the citation datasets, and followed the same dataset splits in [18] for Coauthor Physics, Amazon Computer, and Amazon Photo datasets, for the fair comparison[2].

**Metric**: We used the following metrics for our experiments:

- *Area Under Receiver Operating Characteristics (AUROC)*: AUROC shows the area under the curve where FPR (false positive rate) is in $x$-axis and TPR (true positive rate) is in $y$-axis. It can be interpreted as the probability that a positive example is assigned a higher detection score than a negative example[1]. A perfect detector corresponds to an AUROC score of 100%.

- *Area Under Precision-Prediction Curve (AUPR)*: The PR curve is a graph showing the precision=TP/(TP+FP) and recall=TP/(TP+FN) against each other,and AUPR denotes the area under the precision-recall curve. The ideal case is when Precision is 1 and Recall is 1.

## B.3  Experimental Setup for Out-of-Distribution (OOD) Detection

For OOD detection on semi-supervised node classification, we randomly selected 1-4 categories as OOD categories and trained the models only based on training nodes of the other categories. In this setting, we still trained a model for semi-supervised node classification task, but only part of node categories were not used for training. Hence, we suppose that our model only outputs partial categories (as we don't know the OOD category), see Table 2. For example, Cora dataset, we trained

the model with 80 nodes (20 nodes for each category) with the predictions of 4 categories. Positive ratio is the ratio of out-of-distribution nodes among on all test nodes.

Table 2: Description of datasets and their experimental setup for the OOD detection.

| Dataset | Cora | Citeseer | Pubmed | Co.Physics | Ama.Computer | Ama.Photo |
|---|---|---|---|---|---|---|
| **Number of training categories** | 4 | 3 | 2 | 3 | 5 | 4 |
| **Training nodes** | 80 | 60 | 40 | 60 | 100 | 80 |
| **Test nodes** | 1000 | 1000 | 1000 | 1000 | 1000 | 1000 |
| **Positive ratio** | 38% | 55% | 40.4% | 45.1% | 48.1% | 51.1% |

## B.4 Baseline Setting

In experiment part, we considered 4 baselines. For GCN, we used the same hyper-parameters as [9]. For EDL-GCN, we used the same hyper-parameters as GCN, and replaced softmax layer to activation layer (Relu) with squares loss [17]. For DPN-GCN, we used the same hyper-parameters as GCN, and changed the softmax layer to activation layer (exponential). Note that as we can not generate OOD node, we only used in-distribution loss of (see Eq.12 in [12]) and ignored the OOD part loss. For Drop-GCN, we used the same hyper-parameters as GCN, and set Monte Carlo sampling times $M = 100$, dropout rate equal to 0.5.

## B.5 Time Complexity Analysis

S-BGCN has a similar time complexity with GCN while S-BGCN-T has the double complexity of GCN. For a given network where $|\mathbb{V}|$ is the number of nodes, $|\mathbb{E}|$ is the number of edges, $C$ is the number of dimensions of the input feature vector for every node, $F$ is the number of features for the output layer, and $M$ is Monte Carlo sampling times.

Table 3: Big-O time complexity of our method and baseline GCN.

| Dataset | GCN | S-GCN | S-BGCN | S-BGCN-T | S-BGCN-T-K |
|---|---|---|---|---|---|
| **Time Complexity (Train)** | $O(|\mathbb{E}|CF)$ | $O(|\mathbb{E}|CF)$ | $O(2|\mathbb{E}|CF)$ | $O(2|\mathbb{E}|CF)$ | $O(2|\mathbb{E}|CF)$ |
| **Time Complexity (Test)** | $O(|\mathbb{E}|CF)$ | $O(|\mathbb{E}|CF)$ | $O(M|\mathbb{E}|CF)$ | $O(M|\mathbb{E}|CF)$ | $O(M|\mathbb{E}|CF)$ |

## B.6 Model Setups for semi-supervised node classification

Our models were initialized using Glorot initialization [4] and trained to minimize loss using the Adam SGD optimizer [8]. For the S-BGCN-T-K model, we used the *early stopping strategy* [18] on Coauthor Physics, Amazon Computer and Amazon Photo datasets while *non-early stopping strategy* was used in citation datasets (i.e., Cora, Citeseer and Pubmed). We set bandwidth $\sigma = 1$ for all datasets in GKDE, and set trade off parameters $\lambda_1 = 0.001$ for misclassification detection, $\lambda_1 = 0.1$ for OOD detection and $\lambda_2 = \min(1, t/200)$ (where $t$ is the index of a current training epoch) for both task; other hyperparameter configurations are summarized in Table 4.

For semi-supervised node classification, we used 50 random weight initialization for our models on Citation network datasets. For Coauthor Physics, Amazon Computer and Amazon Photo datasets, we reported the result based on 10 random train/validation/test splits. In both effect of uncertainty on misclassification and the OOD detection, we reported the AUPR and AUROC results in percent averaged over 50 times of randomly chosen 1000 test nodes in all of test sets (except training or validation set) for all models tested on the citation datasets. For S-BGCN-T-K model in these tasks, we used the same hyperparameter configurations as in Table 4, except S-BGCN-T-K Epistemic using 10,000 epochs to obtain the best result.

Table 4: Hyperparameter configurations of S-BGCN-T-K model

| | Cora | Citeseer | Pubmed | Co.Physics | Ama.Computer | Ama.Photo |
|---|---|---|---|---|---|---|
| **Hidden units** | 16 | 16 | 16 | 64 | 64 | 64 |
| **Learning rate** | 0.01 | 0.01 | 0.01 | 0.01 | 0.01 | 0.01 |
| **Dropout** | 0.5 | 0.5 | 0.5 | 0.1 | 0.2 | 0.2 |
| $L_2$ **reg.strength** | 0.0005 | 0.0005 | 0.0005 | 0.001 | 0.0001 | 0.0001 |
| **Monte-Carlo samples** | 100 | 100 | 100 | 100 | 100 | 100 |
| **Max epoch** | 200 | 200 | 200 | 100000 | 100000 | 100000 |

## B.7 Pseudo code for Our Algorithms

---
**Algorithm 1:** S-BGCN-T-K

---
**Input:** $\mathbb{G} = (\mathbb{V}, \mathbb{E}, \mathbf{r})$ and $\mathbf{y}_{\mathbb{L}}$
**Output:** $\mathbf{p}_{\mathbb{V} \setminus \mathbb{L}}, \mathbf{u}_{\mathbb{V} \setminus \mathbb{L}}$

1   $\ell = 0$;
2   Set hyper-parameters $\eta, \lambda_1, \lambda_2$;
3   Initialize the parameters $\gamma, \beta$;
4   Calculate the prior Dirichlet distribution $\text{Dir}(\hat{\alpha})$;
5   Pretrain the teacher network to get $\text{Prob}(\mathbf{y}|\hat{\mathbf{p}})$;
6   **repeat**
7      Forward pass to compute $\boldsymbol{\alpha}$, $\text{Prob}(\mathbf{p}_i|A, \mathbf{r}; \mathcal{G})$ for $i \in \mathbb{V}$;
8      Compute joint probability $\text{Prob}(\mathbf{y}|A, \mathbf{r}; \mathcal{G})$;
9      Backward pass via the chain-rule the calculate the sub-gradient gradient: $g^{(\ell)} = \nabla_\Theta \mathcal{L}(\Theta)$
10     Update parameters using step size $\eta$ via $\Theta^{(\ell+1)} = \Theta^{(\ell)} - \eta \cdot g^{(\ell)}$
11     $\ell = \ell + 1$;
12 **until** *convergence*
13 Calculate $\mathbf{p}_{\mathbb{V} \setminus \mathbb{L}}, \mathbf{u}_{\mathbb{V} \setminus \mathbb{L}}$
14 **return** $p_{\mathbb{V} \setminus \mathbb{L}}, u_{\mathbb{V} \setminus \mathbb{L}}$

---

## B.8 Bayesian Inference with Dropout

The marginalization in Eq.(8) (in main paper) is generally intractable. A dropout technique is used to obtain an approximate solution and use samples from the posterior distribution of models [3]. Hence, we adopted a dropout technique in [2] for variational inference in Bayesian convolutional neural networks where Bernoulli distributions are assumed over the network's weights. This dropout technique allows us to perform probabilistic inference over our Bayesian DL framework using GNNs. For Bayesian inference, we identified a posterior distribution over the network's weights, given the input graph $\mathcal{D}$ and observed labels $\mathbf{y}_{\mathbb{L}}$ by $\text{Prob}(\boldsymbol{\theta}|\mathcal{D})$, where $\boldsymbol{\theta} = \{\mathbf{W}_1, \ldots, \mathbf{W}_L, b_1, ..., b_L\}$, $L$ is the total number of layers and $W_i$ refers to the GNN's weight matrices of dimensions $D_i \times D_{i-1}$, and $b_i$ is a bias vector of dimensions $D_i$ for layer $i = 1, \cdots, L$.

Since the posterior distribution is intractable, we used a **variational inference** to learn $q(\boldsymbol{\theta})$, a distribution over matrices whose columns are randomly set to zero, approximating the intractable posterior by minimizing the Kullback-Leibler (KL)-divergence between this approximated distribution and the full posterior, which is given by:

$$\text{KL}(q(\boldsymbol{\theta})\|\text{Prob}(\boldsymbol{\theta}|\mathcal{D})) \tag{32}$$

We define $\mathbf{W}_i$ in $q(\boldsymbol{\theta})$ by:

$$\mathbf{W}_i = \mathbf{M}_i \text{diag}([z_{ij}]_{j=1}^{D_i}), \quad z_{ij} \sim \text{Bernoulli}(d_i) \text{ for } i = 1, \ldots, L, j = 1, \ldots, D_{i-1} \tag{33}$$

where $\boldsymbol{\gamma} = \{\mathbf{M}_1, \ldots, \mathbf{M}_L, \mathbf{m}_1, \ldots, \mathbf{m}_L\}$ are the variational parameters, $\mathbf{M}_i \in \mathbb{R}^{D_i \times D_{i-1}}, \mathbf{m}_i \in \mathbb{R}^{D_i}$, and $\mathbf{d} = \{d_1, \ldots, d_L\}$ is the dropout probabilities with $z_{ij}$ of Bernoulli distributed random variables. The binary variable $z_{ij} = 0$ corresponds to unit $j$ in layer $i-1$ being dropped out as an input to layer $i$. We can obtain the approximate model of the Gaussian process from [2]. The dropout probabilities, $d_i$'s, can be optimized or fixed [6]. For simplicity, we fixed $d_i$'s in our experiments, as it is beyond the scope of our study. In [2], the minimization of the cross entropy (or square error) loss function is proven to minimize the KL-divergence (see Eq. (32)). Therefore, training the GNN model with stochastic gradient descent enables learning of an approximated distribution of weights, which provides good explainability of data and prevents overfitting.

For the dropout inference, we performed training on a DL model with dropout before every weight layer and dropout at a test time to sample from the approximate posterior (i.e., stochastic forward passes, a.k.a. Monte Carlo dropout; see Eq. (36)). At the test stage, we infer the joint probability by:

$$p(\mathbf{y}|A, \mathbf{r}; \mathcal{D}) = \int \int \text{Prob}(\mathbf{y}|\mathbf{p})\text{Prob}(\mathbf{p}|A, \mathbf{r}; \boldsymbol{\theta})\text{Prob}(\boldsymbol{\theta}|\mathcal{D})d\mathbf{p}d\boldsymbol{\theta}$$

$$\approx \frac{1}{M}\sum_{m=1}^{M}\int \text{Prob}(\mathbf{y}|\mathbf{p})\text{Prob}(\mathbf{p}|A, \mathbf{r}; \boldsymbol{\theta}^{(m)})d\mathbf{p}, \quad \boldsymbol{\theta}^{(m)} \sim q(\boldsymbol{\theta}), \tag{34}$$

where $M$ is Monte Carlo sampling times. We can also infer the Dirichlet parameters $\boldsymbol{\alpha}$ as:

$$\boldsymbol{\alpha} \approx \frac{1}{M}\sum_{m=1}^{M} f(A, \mathbf{r}, \boldsymbol{\theta}^{(m)}), \quad \boldsymbol{\theta}^{(m)} \sim q(\boldsymbol{\theta}). \tag{35}$$

## C   Additional Experimental Results

In addition to the uncertainty analysis in Section 5, we also conducted additional experiments. **First**, we conducted an ablation experiment for each component (such as GKDE, Teacher network, Subjective framework and Bayesian framework) we proposed. **Second**, we provide additional uncertainty visualization results in network node classifications for Citeseer dataset. To clearly understand the effect of different types of uncertainty in classification accuracy and OOD, we used the AUROC and AUPR curves for all types of models considered in this work.

### C.1   Ablation Experiments

We conducted an additional experiments in order to clearly demonstrate the contributions of the key technical components, including a teacher Network, Graph kernel Dirichlet Estimation (GKDE) and subjective Bayesian framework. The key findings obtained from this experiment are: (1) The teacher Network can further improve node classification accuracy (i.e., 0.2% - 1.5% increase, as shown in Table 5); and (2) GKDE (Graph-Based Kernel Dirichlet Distribution Estimation) using the uncertainty estimates can enhance OOD detection (i.e., 4% - 30% increase, as shown in Table 6).

Table 5: Ablation experiment on AUROC and AUPR for the Misclassification Detection.

| Data | Model | AUROC | | | | | AUPR | | | | | |
|---|---|---|---|---|---|---|---|---|---|---|---|---|
| | | Va. | Dis. | Al. | Ep. | En. | Va. | Dis. | Al. | Ep. | En. | Acc |
| Cora | S-BGCN-T-K | 70.6 | 82.4 | 75.3 | 68.8 | 77.7 | 90.3 | **95.4** | 92.4 | 87.8 | 93.4 | 82.0 |
| | S-BGCN-T | 70.8 | **82.5** | 75.3 | 68.9 | 77.8 | 90.4 | **95.4** | 92.6 | 88.0 | 93.4 | **82.2** |
| | S-BGCN | 69.8 | 81.4 | 73.9 | 66.7 | 76.9 | 89.4 | 94.3 | 92.3 | 88.0 | 93.1 | 81.2 |
| | S-GCN | 70.2 | 81.5 | - | - | 76.9 | 90.0 | 94.6 | - | - | 93.6 | 81.5 |
| Citeseer | S-BGCN-T-K | 65.4 | **74.0** | 67.2 | 60.7 | 70.0 | 79.8 | **85.6** | 82.2 | 75.2 | 83.5 | 71.0 |
| | S-BGCN-T | 65.4 | 73.9 | 67.1 | 60.7 | 70.1 | 79.6 | 85.5 | 82.1 | 75.2 | 83.5 | **71.3** |
| | S-BGCN | 63.9 | 72.1 | 66.1 | 58.9 | 69.2 | 78.4 | 83.8 | 80.6 | 75.6 | 82.3 | 70.6 |
| | S-GCN | 64.9 | 71.9 | - | - | 69.4 | 79.5 | 84.2 | - | - | 82.5 | 71.0 |
| Pubmed | S-BGCN-T-K | 63.1 | **69.9** | 66.5 | 65.3 | 68.1 | 85.6 | 90.8 | 88.8 | 86.1 | 89.2 | **79.3** |
| | S-BGCN-T | 63.2 | **69.9** | 66.6 | 65.3 | 64.8 | 85.6 | **90.9** | 88.9 | 86.0 | 89.3 | 79.2 |
| | S-BGCN | 62.7 | 68.1 | 66.1 | 64.4 | 68.0 | 85.4 | 90.5 | 88.6 | 85.6 | 89.2 | 78.8 |
| | S-GCN | 62.9 | 69.5 | - | - | 68.0 | 85.3 | 90.4 | - | - | 89.2 | 79.1 |
| Amazon Photo | S-BGCN-T-K | 66.0 | 89.3 | 83.0 | 83.4 | 83.2 | 95.4 | **98.9** | 98.4 | 98.1 | 98.4 | 92.0 |
| | S-BGCN-T | 66.1 | 89.3 | 83.1 | 83.5 | 83.3 | 95.6 | 99.0 | 98.4 | 98.2 | 98.4 | **92.3** |
| | S-BGCN | 68.6 | **93.6** | 90.6 | 83.6 | 90.6 | 90.4 | 98.1 | 97.3 | 95.8 | 97.3 | 81.0 |
| | S-GCN | - | - | - | - | 86.7 | - | - | - | - | - | 98.4 |
| Amazon Computer | S-BGCN-T-K | 65.0 | 87.8 | 83.3 | 79.6 | 83.6 | 89.4 | 96.3 | 95.0 | 94.2 | 95.0 | 84.0 |
| | S-BGCN-T | 65.2 | 88.0 | 83.4 | 79.7 | 83.6 | 89.4 | **96.5** | 95.0 | 94.5 | 95.1 | **84.1** |
| | S-BGCN | 63.7 | **89.1** | 84.3 | 76.1 | 84.4 | 84.9 | 95.7 | 93.9 | 91.4 | 93.9 | 76.1 |
| | S-GCN | - | - | - | - | 81.5 | - | - | - | - | - | 95.2 |
| Coauthor Physics | S-BGCN-T-K | 80.2 | 91.4 | 87.5 | 81.7 | 87.6 | 98.3 | **99.4** | 99.0 | 98.4 | 98.9 | 93.0 |
| | S-BGCN-T | 80.4 | **91.5** | 87.6 | 81.7 | 87.6 | 98.3 | **99.4** | 99.0 | 98.6 | 99.0 | **93.2** |
| | S-BGCN | 79.6 | 90.5 | 86.3 | 81.2 | 86.4 | 98.0 | 99.2 | 98.8 | 98.3 | 98.8 | 92.9 |
| | S-GCN | 89.1 | 89.0 | - | - | 89.2 | 99.0 | 99.0 | - | - | 99.0 | 92.9 |

Va.: Vacuity, Dis.: Dissonance, Al.: Aleatoric, Ep.: Epistemic, En.: Entropy

### C.2   Experiment based on GAT model

We also conducted the semi-supervised node classification based on GAT model [20]).Model setup: The S-BGAT-T-K model has two dropout probabilities, which are a dropout on features and a dropout

Table 6: Ablation experiment on AUROC and AUPR for the OOD Detection.

| Data | Model | AUROC | | | | | AUPR | | | | |
|---|---|---|---|---|---|---|---|---|---|---|---|
| | | Va. | Dis. | Al. | Ep. | En. | Va. | Dis. | Al. | Ep. | En. |
| Cora | S-BGCN-T-K | **87.6** | 75.5 | 85.5 | 70.8 | 84.8 | **78.4** | 49.0 | 75.3 | 44.5 | 73.1 |
| | S-BGCN-T | 84.5 | 81.2 | 83.5 | 71.8 | 83.5 | 74.4 | 53.4 | 75.8 | 46.8 | 71.7 |
| | S-BGCN | 76.3 | 79.3 | 81.5 | 70.5 | 80.6 | 61.3 | 55.8 | 68.9 | 44.2 | 65.3 |
| | S-GCN | 75.0 | 78.2 | - | - | 79.4 | 60.1 | 54.5 | - | - | 65.3 |
| Citeseer | S-BGCN-T-K | **84.8** | 55.2 | 78.4 | 55.1 | 74.0 | **86.8** | 54.1 | 80.8 | 55.8 | 74.0 |
| | S-BGCN-T | 78.6 | 59.6 | 73.9 | 56.1 | 69.3 | 79.8 | 57.4 | 76.4 | 57.8 | 69.3 |
| | S-BGCN | 72.7 | 63.9 | 72.4 | 61.4 | 70.5 | 73.0 | 62.7 | 74.5 | 60.8 | 71.6 |
| | SGCN | 72.0 | 62.8 | - | - | 70.0 | 71.4 | 61.3 | - | - | 70.5 |
| Pubmed | S-BGCN-T-K | **74.6** | 67.9 | 71.8 | 59.2 | 72.2 | **69.6** | 52.9 | 63.6 | 44.0 | 56.5 |
| | S-BGCN-T | 71.8 | 68.6 | 70.0 | 60.1 | 70.8 | 65.7 | 53.9 | 61.8 | 46.0 | 55.1 |
| | S-BGCN | 70.8 | 68.2 | 70.3 | 60.8 | 68.0 | 65.4 | 53.2 | 62.8 | 46.7 | 55.4 |
| | S-GCN | 71.4 | 68.8 | - | - | 69.7 | 66.3 | 54.9 | - | - | 57.5 |
| Amazon Photo | S-BGCN-T-K | **93.4** | 76.4 | 91.4 | 32.2 | 91.4 | **94.8** | 68.0 | 92.3 | 42.3 | 92.5 |
| | S-BGCN-T | 64.0 | 77.5 | 79.9 | 52.6 | 79.8 | 67.0 | 75.3 | 82.0 | 53.7 | 81.9 |
| | S-BGCN | 63.0 | 76.6 | 79.8 | 52.7 | 79.7 | 66.5 | 75.1 | 82.1 | 53.9 | 81.7 |
| | S-GCN | 64.0 | 77.1 | - | - | 79.6 | 67.0 | 74.9 | - | - | 81.6 |
| Amazon Computer | S-BGCN-T-K | **82.3** | 76.6 | 80.9 | 55.4 | 80.9 | **70.5** | 52.8 | 60.9 | 35.9 | 60.6 |
| | S-BGCN-T | 53.7 | 70.5 | 70.4 | 69.9 | 70.1 | 33.6 | 43.9 | 46.0 | 46.8 | 45.9 |
| | S-BGCN | 56.9 | 75.3 | 74.1 | 73.7 | 74.1 | 33.7 | 46.2 | 48.3 | 45.6 | 48.3 |
| | S-GCN | 56.9 | 75.3 | - | - | 74.2 | 33.7 | 46.2 | - | - | 48.3 |
| Coauthor Physics | S-BGCN-T-K | **91.3** | 87.6 | 89.7 | 61.8 | 89.8 | **72.2** | 56.6 | 68.1 | 25.9 | 67.9 |
| | S-BGCN-T | 88.7 | 86.0 | 87.9 | 70.2 | 87.8 | 67.4 | 51.9 | 64.6 | 29.4 | 62.4 |
| | S-BGCN | 89.1 | 87.1 | 89.5 | 78.3 | 89.5 | 66.1 | 49.2 | 64.6 | 35.6 | 64.3 |
| | S-GCN | 89.1 | 87.0 | - | - | 89.4 | -66.2 | 49.2 | - | - | 64.3 |

Va.: Vacuity, Dis.: Dissonance, Al.: Aleatoric, Ep.: Epistemic, D.En.: Differential Entropy, En.: Entropy

on attention coefficients, as shown in Table 7. We changed the dropout on attention coefficients to 0.4 at the test stage and set trade off parameters $\lambda = \min(1, t/50)$, using the same early stopping strategy [20]. The result are shown in Table 8.

Table 7: Hyper-parameters of S-BGAT-T-K model

| | Cora | Citeseer | Pubmed |
|---|---|---|---|
| **Hidden units** | 64 | 64 | 64 |
| **Learning rate** | 0.01 | 0.01 | 0.01 |
| **Dropout** | 0.6/0.6 | 0.6/0.6 | 0.6/0.6 |
| $L_2$ **reg.strength** | 0.0005 | 0.0005 | 0.001 |
| **Monte-Carlo samples** | 100 | 100 | 100 |
| **Max epoch** | 100000 | 100000 | 100000 |

Table 8: Semi-supervised node classification accuracy based on GAT

| | Cora | Citeseer | Pubmed |
|---|---|---|---|
| **GAT** | $83.0 \pm 0.7$ | $72.5 \pm 0.7$ | $79.0 \pm 0.3$ |
| **GAT-Drop** | $82.8 \pm 0.8$ | $72.6 \pm 0.7$ | $79.0 \pm 0.3$ |
| **S-GAT** | $83.0 \pm 0.7$ | $72.6 \pm 0.6$ | $79.0 \pm 0.3$ |
| **S-BGAT** | $82.9 \pm 0.7$ | $72.4 \pm 0.7$ | $78.9 \pm 0.3$ |
| **S-BGAT-T** | $83.7 \pm 0.6$ | $\mathbf{73.2 \pm 0.5}$ | $79.1 \pm 0.2$ |
| **S-BGAT-T-K** | $\mathbf{83.8 \pm 0.7}$ | $73.0 \pm 0.7$ | $\mathbf{79.1 \pm 0.2}$ |

### C.3 Misclassification Detection

For Amazon Photo, Amazon Computer and Coauthor Physics dataset, the misclassification detection results are shown in Tabel 9.

### C.4 Graph Embedding Representations of Different Uncertainty Types

To better understand different uncertainty types, we used $t$-SNE ($t$-Distributed Stochastic Neighbor Embedding [11]) to represent the computed feature representations of a pre-trained BGCN-T model's first hidden layer on the Cora dataset and the Citeseer dataset.

Table 9: AUROC and AUPR for the Misclassification Detection.

| Data | Model | AUROC | | | | | AUPR | | | | | Acc |
|---|---|---|---|---|---|---|---|---|---|---|---|---|
| | | Va. | Dis. | Al. | Ep. | En. | Va. | Dis. | Al. | Ep. | En. | |
| Amazon Photo | S-BGCN-T-K | 66.0 | **89.3** | 83.0 | 83.4 | 83.2 | 95.4 | **98.9** | 98.4 | 98.1 | 98.4 | **92.0** |
| | EDL-GCN | 65.1 | 88.5 | - | - | 82.2 | 94.6 | 98.1 | - | - | 98.0 | 91.2 |
| | DPN-GCN | - | - | 81.8 | 80.8 | 81.3 | - | - | 98.1 | 98.0 | 98.0 | **92.0** |
| | Drop-GCN | - | - | 84.5 | 84.4 | 84.6 | - | - | 98.2 | 98.1 | 98.2 | 91.3 |
| | GCN | - | - | - | - | 86.8 | - | - | - | - | 98.5 | 91.2 |
| Amazon Computer | S-BGCN-T-K | 65.0 | **87.8** | 83.3 | 79.6 | 83.6 | 89.4 | **96.3** | 95.0 | 94.2 | 95.0 | 84.0 |
| | EDL-GCN | 64.1 | 86.5 | - | - | 82.2 | 93.6 | 97.1 | - | - | 97.0 | 79.7 |
| | DPN-GCN | - | - | 76.8 | 76.0 | 76.3 | - | - | 94.5 | 94.3 | 94.4 | **84.8** |
| | Drop-GCN | - | - | 79.1 | 75.9 | 79.2 | - | - | 95.1 | 94.5 | 95.1 | 79.6 |
| | GCN | - | - | - | - | 81.7 | - | - | - | - | 95.4 | 82.6 |
| Coauthor Physics | S-BGCN-T-K | 80.2 | **91.4** | 87.5 | 81.7 | 87.6 | 98.3 | **99.4** | 99.0 | 98.4 | 98.9 | **93.0** |
| | EDL-GCN | 78.8 | 89.5 | - | - | 86.2 | 96.6 | 97.2 | - | - | 97.0 | 92.7 |
| | DPN-GCN | - | - | 87.0 | 86.4 | 86.8 | - | - | 99.1 | 99.0 | 99.0 | 92.5 |
| | Drop-GCN | - | - | 87.6 | 84.1 | 87.7 | - | - | 98.9 | 98.6 | 98.9 | 93.0 |
| | GCN | - | - | - | - | 88.7 | - | - | - | - | 99.0 | 92.8 |

Va.: Vacuity, Dis.: Dissonance, Al.: Aleatoric, Ep.: Epistemic, En.: Entropy

Figure 1: Graph embedding representations of the Cora dataset for classes and the extent of uncertainty: (a) shows the representation of seven different classes; (b) shows our model prediction; and (c)-(f) present the extent of uncertainty for respective uncertainty types, including vacuity, dissonance, aleatoric, epistemic.

Figure 2: Graph embedding representations of the Citeseer dataset for classes and the extent of uncertainty: (a) shows the representation of seven different classes, (b) shows our model prediction and (c)-(f) present the extent of uncertainty for respective uncertainty types, including vacuity, dissonance, and aleatoric uncertainty, respectively.

**Seven Classes on Cora Dataset**: In Figure 1, (a) shows the representation of seven different classes, (b) shows our model prediction and (c)-(f) present the extent of uncertainty for respective uncertainty types, including vacuity, dissonance, and aleatoric uncertainty, respectively.

(a) PR curves on Cora     (b) PR curves on Citeseer     (c) PR curves on Pubmed

Figure 3: PR curves of misclassification detection for S-BGCN-T-K and other baselines, GCN-Drop and GCN.

(a) Amazon Photo     (b) Amazon Computers     (c) Coauthor Physics

Figure 4: PR cuves of OOD detection for S-BGCN-T-K with uncertainties.

**Six Classes on Citeseer Dataset**: In Figure 2 (a), a node's color denotes a class on the Citeseer dataset where 6 different classes are shown in different colors. Figure 2 (b) is our prediction result.

For Figures 2 (c)-(f), the extent of uncertainty is presented where a blue color refers to the lowest uncertainty (i.e., minimum uncertainty) while a red color indicates the highest uncertainty (i.e., maximum uncertainty) based on the presented color bar. To examine the trends of the extent of uncertainty depending on either training nodes or test nodes, we draw training nodes as bigger circles than test nodes. Overall we notice that most training nodes (shown as bigger circles) have low uncertainty (i.e., blue), which is reasonable because the training nodes are the ones that are already observed. Now we discuss the extent of uncertainty under each uncertainty type.

**Vacuity**: In Figure 2 (c), although most training nodes show low uncertainty, we observe majority of test nodes in the mid cluster show high uncertainty as appeared in red.

**Dissonance**: In Figure 2 (d), similar to vacuity, training nodes have low uncertainty. But unlike vacuity, test nodes are much less uncertain. Recall that dissonance represents the degree of conflicting evidence (i.e., discrepancy between each class probability). However, in this dataset, we observe a fairly low level of dissonance and the obvious outperformance of Dissonance in node classification prediction.

**Aleatoric uncertainty**: In Figure 2 (e), a lot of nodes show high uncertainty with larger than 0.5 except a small amount of training nodes with low uncertainty.

**Epistemic uncertainty**: In Figure 2 (f), most nodes show very low epistemic uncertainty values because uncertainty derived from model parameters can disappear as they are trained well.

## C.5 PR and ROC Curves

**AUPR for the OOD Detection**: Figure 4 shows the AUPRC for the OOD detection when S-BGCN-T-K is used to detect OOD in which test nodes are considered based on their high uncertainty level, given a different uncertainty type, such as vacuity, dissonance, aleatoric, epistemic, or entropy (or total uncertainty). Also to check the performance of the proposed models with a baseline model, we added S-BGCN-T-K with test nodes randomly selected (i.e., Random).

| (a) Amazon Photo | (b) Amazon Computers | (c) Coauthor Physics |

Figure 5: ROC curves of OOD detection for S-BGCN-T-K with uncertainties.

Obviously, in Random baseline, precision was not sensitive to increasing recall while in S-BGCN-T-K (with test nodes being selected based on high uncertainty) precision decreases as recall increases. But although most S-BGCN-T-K models with various uncertainty types used to select test nodes shows sensitive precision to increasing recall (i.e., proving uncertainty being an indicator of improving OOD detection). In addition, unlike AUPR in misclassification detection, which showed the best performance in S-BGCN-T-K Dissonance (see Figure 3), S-BGCN-T-K Dissonance showed the second worst performance among the proposed S-BGCN-T-K models with other uncertainty types. This means that less conflicting information does not help OOD detection. On the other hand, overall we observed Vacuity performs the best among all. From this finding, we can claim that to improve OOD detection, less information with a high vacuity value can help boost the accuracy of the OOD detection.

**AUROC for the OOD Detection**: First, we investigated the performance of our proposed S-BGCN-T-K models when test nodes are selected based on seven different criteria (i.e., uncertainty measures). For AUROC in Figure 5, we observed much better performance in most S-BGCN-T-K models with all uncertainty types except epistemic uncertainty.

## C.6 Analysis for Epistemic Uncertainty in OOD Detection

Although epistemic uncertainty is known to be effective to improve OOD detection [3, 7] in computer vision applications, our results demonstrate it is less effective than our vacuity-based approach. One potential reason is that the previous success of epistemic in computer vision applications are only applied in supervised learning, but they are not sufficiently validated in semi-supervised learning.

To back up our conclusion, designe a image classification experiment based on MC-Drop[3] method to do the following experiment: 1) supervised learning on MNIST dataset with 50 labeled images; 2) semi-supervised learning (SSL) on MNIST dataset with 50 labeled images and 49950 unlabeled images, while there are 50% OOD images (24975 FashionMNIST images) in unlabeled set. For both experiment, we test the epistemic uncertainty on 49950 unlabeled set (50% In-distribution (ID) images and 50% OOD images). We conduct the experiment the experiment based on three popular SSL methods, VAT [14], Mean Teacher [19] and pseudo label [10]. Table 10 shows the average

Table 10: Epistemic uncertainty for semi-supervised image classification.

| Epistemic | Supervised | VAT | Mean Teacher | Pseudo Label |
|---|---|---|---|---|
| **In-Distribution** | 0.140 | **0.116** | **0.105** | **0.041** |
| **Out-of-Distribution** | **0.249** | 0.049 | 0.076 | 0.020 |

epistemic uncertainty value for in-distribution samples and OOD samples. The result shows the same pattern with [7, 6] in a supervised setting, but an opposite pattern in a semi-supervised setting that low epistemic of OOD samples, which is less effective top detect OOD. Note that the SSL setting is similar to our semi-supervised node classification setting, which feed the unlabeled sample to train the model.

## C.7 Compare with Bayesian GCN baseline

Compare with a (Bayesian) GCN baseline, Dropout+DropEdge [15]. As shown in the table 11 below, our proposed method performed better than Dropout+DropEdge on the Cora and Citeer datasets for misclassificaiton detection. A similar trend was observed for OOD detection.

Table 11: Compare with DropEdge on Misclassification Detection .

| Dataset | Model | AUROC | | | | | AUPR | | | | |
|---------|-------|-------|------|------|------|------|------|------|------|------|------|
|         |       | Va.   | Dis. | Al.  | Ep.  | En.  | Va.  | Dis. | Al.  | Ep.  | En.  |
| Cora    | S-BGCN-T-K | 70.6 | **82.4** | 75.3 | 68.8 | 77.7 | 90.3 | **95.4** | 92.4 | 87.8 | 93.4 |
|         | DropEdge   | -    | -        | 76.6 | 56.1 | 76.6 | -    | -        | 93.2 | 85.4 | 93.2 |
| Citeseer | S-BGCN-T-K | 65.4 | **74.0** | 67.2 | 60.7 | 70.0 | 79.8 | **85.6** | 82.2 | 75.2 | 83.5 |
|          | DropEdge   | -    | -        | 71.1 | 51.2 | 71.1 | -    | -        | 84.0 | 70.3 | 84.0 |

Va.: Vacuity, Dis.: Dissonance, Al.: Aleatoric, Ep.: Epistemic, En.: Entropy

# D Derivations for Joint Probability and KL Divergence

## D.1 Joint Probability

At the test stage, we infer the joint probability by:

$$
\begin{aligned}
p(\mathbf{y}|A, \mathbf{r}; \mathcal{G}) &= \int \int \text{Prob}(\mathbf{y}|\mathbf{p})\text{Prob}(\mathbf{p}|A, \mathbf{r}; \boldsymbol{\theta})\text{Prob}(\boldsymbol{\theta}|\mathcal{G})d\mathbf{p}d\boldsymbol{\theta} \\
&\approx \int \int \text{Prob}(\mathbf{y}|\mathbf{p})\text{Prob}(\mathbf{p}|A, \mathbf{r}; \boldsymbol{\theta})q(\boldsymbol{\theta})d\mathbf{p}d\theta \\
&\approx \frac{1}{M}\sum_{m=1}^{M}\int \text{Prob}(\mathbf{y}|\mathbf{p})\text{Prob}(\mathbf{p}|A, \mathbf{r}; \boldsymbol{\theta}^{(m)})d\mathbf{p}, \quad \boldsymbol{\theta}^{(m)} \sim q(\boldsymbol{\theta}) \\
&\approx \frac{1}{M}\sum_{m=1}^{M}\int \sum_{i=1}^{N}\text{Prob}(\mathbf{y}_i|\mathbf{p}_i)\text{Prob}(\mathbf{p}_i|A, \mathbf{r}; \boldsymbol{\theta}^{(m)})d\mathbf{p}_i, \quad \boldsymbol{\theta}^{(m)} \sim q(\boldsymbol{\theta}) \\
&\approx \frac{1}{M}\sum_{m=1}^{M}\sum_{i=1}^{N}\int \text{Prob}(\mathbf{y}_i|\mathbf{p}_i)\text{Prob}(\mathbf{p}_i|A, \mathbf{r}; \boldsymbol{\theta}^{(m)})d\mathbf{p}_i, \quad \boldsymbol{\theta}^{(m)} \sim q(\boldsymbol{\theta}) \\
&\approx \frac{1}{M}\sum_{m=1}^{M}\prod_{i=1}^{N}\int \text{Prob}(\mathbf{y}_i|\mathbf{p}_i)\text{Dir}(\mathbf{p}_i|\boldsymbol{\alpha}_i^{(m)})d\mathbf{p}_i, \quad \boldsymbol{\alpha}^{(m)} = f(A, \mathbf{r}, \boldsymbol{\theta}^{(m)}), q \quad \boldsymbol{\theta}^{(m)} \sim q(\boldsymbol{\theta}),
\end{aligned}
$$

where the posterior over class label $p$ will be given by the mean of the Dirichlet:

$$
\text{Prob}(y_i = p|\boldsymbol{\theta}^{(m)}) = \int \text{Prob}(y_i = p|\mathbf{p}_i)\text{Prob}(\mathbf{p}_i|A, \mathbf{r}; \boldsymbol{\theta}^{(m)})d\mathbf{p}_i = \frac{\alpha_{ip}^{(m)}}{\sum_{k=1}^{K}\alpha_{ik}^{(m)}}.
$$

The probabilistic form for a specific node $i$ by using marginal probability,

$$
\begin{aligned}
\text{Prob}(\mathbf{y}_i|A,\mathbf{r};\mathcal{G}) &= \sum_{y\backslash y_i} \text{Prob}(\mathbf{y}|A,\mathbf{r};\mathcal{G}) \\
&= \sum_{y\backslash y_i} \int\int \prod_{j=1}^N \text{Prob}(\mathbf{y}_j|\mathbf{p}_j)\text{Prob}(\mathbf{p}_j|A,\mathbf{r};\boldsymbol{\theta})\text{Prob}(\boldsymbol{\theta}|\mathcal{G})d\mathbf{p}d\boldsymbol{\theta} \\
&\approx \sum_{y\backslash y_i} \int\int \prod_{j=1}^N \text{Prob}(\mathbf{y}_j|\mathbf{p}_j)\text{Prob}(\mathbf{p}_j|A,\mathbf{r};\boldsymbol{\theta})q(\boldsymbol{\theta})d\mathbf{p}d\boldsymbol{\theta} \\
&\approx \sum_{m=1}^M \sum_{y\backslash y_i} \int \prod_{j=1}^N \text{Prob}(\mathbf{y}_j|\mathbf{p}_j)\text{Prob}(\mathbf{p}_j|A,\mathbf{r};\boldsymbol{\theta}^{(m)})d\mathbf{p}, \quad \boldsymbol{\theta}^{(m)}\sim q(\boldsymbol{\theta}) \\
&\approx \sum_{m=1}^M \Big[\sum_{y\backslash y_i} \int \prod_{j=1}^N \text{Prob}(\mathbf{y}_j|\mathbf{p}_j)\text{Prob}(\mathbf{p}_j|A,\mathbf{r};\boldsymbol{\theta}^{(m)})d\mathbf{p}_j\Big], \quad \boldsymbol{\theta}^{(m)}\sim q(\boldsymbol{\theta}) \\
&\approx \sum_{m=1}^M \Big[\sum_{y\backslash y_i} \prod_{j=1,j\neq i}^N \text{Prob}(\mathbf{y}_j|A,\mathbf{r}_j;\boldsymbol{\theta}^{(m)})\Big]\text{Prob}(\mathbf{y}_i|A,\mathbf{r};\boldsymbol{\theta}^{(m)}), \quad \boldsymbol{\theta}^{(m)}\sim q(\boldsymbol{\theta}) \\
&\approx \sum_{m=1}^M \int \text{Prob}(\mathbf{y}_i|\mathbf{p}_i)\text{Prob}(\mathbf{p}_i|A,\mathbf{r};\boldsymbol{\theta}^{(m)})d\mathbf{p}_i, \quad \boldsymbol{\theta}^{(m)}\sim q(\boldsymbol{\theta}).
\end{aligned}
$$

To be specific, the probability of label $p$ is,

$$
\text{Prob}(y_i=p|A,\mathbf{r};\mathcal{G}) \approx \frac{1}{M}\sum_{m=1}^M \frac{\alpha_{ip}^{(m)}}{\sum_{k=1}^K \alpha_{ik}^{(m)}}, \quad \boldsymbol{\alpha}^{(m)}=f(A,\mathbf{r},\boldsymbol{\theta}^{(m)}), \quad \boldsymbol{\theta}^{(m)}\sim q(\boldsymbol{\theta}).
$$

### D.2 KL-Divergence

KL-divergence between $\text{Prob}(\mathbf{y}|\mathbf{r};\boldsymbol{\gamma},\mathcal{G})$ and $\text{Prob}(\mathbf{y}|\hat{\mathbf{p}})$ is given by

$$
\begin{aligned}
\text{KL}[\text{Prob}(\mathbf{y}|A,\mathbf{r};\mathcal{G})||\text{Prob}(\mathbf{y}|\hat{\mathbf{p}}))] &= \mathbb{E}_{\text{Prob}(\mathbf{y}|A,\mathbf{r};\mathcal{G})}\Big[\log\frac{\text{Prob}(\mathbf{y}|A,\mathbf{r};\mathcal{G})}{\text{Prob}(\mathbf{y}|\hat{\mathbf{p}})}\Big] \\
&\approx \mathbb{E}_{\text{Prob}(\mathbf{y}|A,\mathbf{r};\mathcal{G})}\Big[\log\frac{\prod_{i=1}^N \text{Prob}(\mathbf{y}_i|A,\mathbf{r};\mathcal{G})}{\prod_{i=1}^N \text{Prob}(\mathbf{y}|\hat{\mathbf{p}})}\Big] \\
&\approx \mathbb{E}_{\text{Prob}(\mathbf{y}|A,\mathbf{r};\mathcal{G})}\Big[\sum_{i=1}^N \log\frac{\text{Prob}(\mathbf{y}_i|A,\mathbf{r};\mathcal{G})}{\text{Prob}(\mathbf{y}|\hat{\mathbf{p}})}\Big] \\
&\approx \sum_{i=1}^N \mathbb{E}_{\text{Prob}(\mathbf{y}|A,\mathbf{r};\mathcal{G})}\Big[\log\frac{\text{Prob}(\mathbf{y}_i|A,\mathbf{r};\mathcal{G})}{\text{Prob}(\mathbf{y}|\hat{\mathbf{p}})}\Big] \\
&\approx \sum_{i=1}^N\sum_{j=1}^K \text{Prob}(y_i=j|A,\mathbf{r};\mathcal{G})\Big(\log\frac{\text{Prob}(y_i=j|A,\mathbf{r};\mathcal{G})}{\text{Prob}(y_i=j|\hat{\mathbf{p}})}\Big)
\end{aligned}
$$

The KL divergence between two Dirichlet distributions $\text{Dir}(\alpha)$ and $\text{Dir}(\hat{\alpha})$ can be obtained in closed form as,

$$
\text{KL}[\text{Dir}(\alpha)||\text{Dir}(\hat{\alpha})] = \ln\Gamma(S) - \ln\Gamma(\hat{S}) + \sum_{c=1}^K \big(\ln\Gamma(\hat{\alpha}_c)-\ln\Gamma(\alpha_c)\big) + \sum_{c=1}^K (\alpha_c-\hat{\alpha}_c)(\psi(\alpha_c)-\psi(S)),
$$

where $S=\sum_{c=1}^K \alpha_c$ and $\hat{S}=\sum_{c=1}^K \hat{\alpha}_c$.

## Footnotes

[1]KDD Cup 2016 Dataset: Online Available at `https://kddcup2016.azurewebsites.net/`

[2]The source code and datasets are accessible at https://github.com/zxj32/uncertainty-GNN