[Reviews · NeurIPS 2020]

Review 1

Summary and Contributions: This paper proposed a multi-source uncertainty framework of GNNs which models various types of uncertainty. The authors study multiple types of uncertainty in deep learning and belief/evidence theory domain for node classification predictions. They validated their presented model with existing benchmarks on six real network datasets for OOD detention and misclassification detection.

Strengths: Pre-existing work in GNNs have not consider the concept of uncertainty associated with probabilities, which can minimize the risk of misclassification. One of the good contributions in the paper is that the authors theoretically analyzed the relationships between different types of uncertainties.

Weaknesses: Even though this is an interesting setting and the technical solutions presented in the paper look reasonable, the idea seems to be pretty incremental as it stacks multiple existing techniques without many innovations. For me, it’s a bit hard to advocate acceptance yet.

Correctness: The described technical details seem to be correct, yet it’s a bit hard to understand the proposed methodology, which needs to be cleared for improvement.

Clarity: This paper is written well in general.

Relation to Prior Work: This paper thoroughly analyzed the differences in multiple types of uncertainties introduced in pre-existing work.

Reproducibility: Yes

Additional Feedback: Can authors provide more detailed rationales for using Dirichlet distribution to model a probability distribution for class probabilities? [Additional comments] I carefully read through authors’ feedback and came to realize the proposed framework can be significantly useful in terms of modeling various types of uncertainty and the experimental results and analysis are reasonable. So, I increased my score from “marginally below the acceptance threshold” to “marginally above the acceptance threshold”.


Review 2

Summary and Contributions: The paper formulates GNNs to output the parameters of Dirichlet distribution over class probabilities instead of directly predicting class probabilities. The goal is to calculate additional uncertainty metrics from belief theory domain, namely dissonance and vacuity. The training is further guided by a Dirichlet prior constructed based on a graph-based kernel estimation, and a GNN trained with only the classification loss. The experiments on semi-supervised node classification on 6 datasets show that dissonance and vacuity can be good scores for detecting misclassification and OoD, respectively.

Strengths: The paper tries to extend the notion of uncertainty estimates in semi-supervised learning on graph data and proposes using dissonance and vacuity as additional uncertainty measures. It models the parameters of a Dirichlet distribution by GNNs to be able to employ dissonance and vacuity scores. Dissonance and vacuity show advantages for in-distribution misclassification detection and OoD detection on several datasets. The paper shows the connection of entropy, epistemic uncertainty, aleatoric uncertainty, dissonance and vacuity under its framework. As the training is guided by multiple components including the teacher GNN and the constructed Dirichlet prior based on the shortest paths from the labeled nodes in the graph, the ablation studies appropriately examine the effect of each component of the framework.

Weaknesses: The scope of the paper is limited to the semi-supervised setting. In the previous works on CV, epistemic uncertainty or entropy have been used for uncertainty estimation/OoD detection. This paper suggests that these uncertainty estimates would not work in semi-supervised settings and provides an ablation study on image classification for demonstrating the claim. On the other hand, the paper does not investigate the performance change with different number of labeled nodes on the graph data which is the focus of the paper and may be more relevant. Some stronger (Bayesian) GCN baselines can be investigated in terms of their uncertainty quantification like Dropout+DropEdge in Rong et. al, 2019.

Correctness: Overall, the method and empirical comparisons seem sound. I think the loss should include an additional KL term for theta because of the Bayesian treatment of it which is not mentioned in the paper.

Clarity: The paper is generally well-written, but there are some minor typos and grammatical issues.

Relation to Prior Work: The paper discusses some related works for uncertainty estimation in BNNs and uncertainty reasoning in belief theory domain and motivates its contributions and relates them to the previous works. The paper may want to discuss more and differentiate between the different approaches for uncertainty estimation in graphs, including dropout-based techniques, and methods that consider uncertainty in graph structure and discuss why the latter ones have not been investigated in this work.

Reproducibility: Yes

Additional Feedback: --Update-- Based on the additional results in the rebuttal that are encouraging, specifically the comparison with DropEdge and providing the performance change with different number of labeled nodes, I have increased my score. However, I still believe that instead of a semi-supervised image classification experiment, the paper needs to investigate a supervised graph experiment setup to demonstrate the claims in section 6.3 about performance differences in supervised vs semi-supervised settings. Specifically, epistemic being worse than all other uncertainty measures (for a fixed method) for OOD detection seems to contradict previous papers in the OOD detection field. Also, I think the loss is missing an L2 regularization on theta that should appear from the KL term of the Bayesian treatment of it.


Review 3

Summary and Contributions: The authors propose an uncertainty framework for GNNs that incorporates several components of uncertainty in data. They have a theoretic analysis of these sources of uncertainty and relate them to each other. They also develop the Graph-based Kernel Dirichlet distribution Estimation (GKDE) model. They train this model and compare it to other popular GNNs, such as GCN, by running many experiments.

Strengths: Figure 1 is an excellent illustration of the different types of scenarios. There are a lot of definitions are variables to digest so this figure really helps LINK the variables to the reader’s intuition. The math and explanations are very straightforward and clear. The authors did an excellent job presenting their work. The questions and answers in the experiment section are very helpful and useful to read over before jumping into the tables. The experiments are thorough and cover many datasets and models.

Weaknesses: The results in Table 1 are good but not amazing. GCN still performs relatively well compared to the authors’ methods. Otherwise, the paper is very solid.

Correctness: Everything seems correct. There are a few things that I was confused about which I mentioned below.

Clarity: Overall the paper is very clear. The authors did an excellent job. Equation 5 - I am confused on a few things. The notation P(y|x; theta) is confusing because the semicolon implies that theta is a vector and not a random vector, however, the conditional distribution of theta is given P(theta|G). So what is the point of the semicolon? Also, there is a typo in Equation 5 I think because the entropy term is not defined correctly. It should be H ( E_{P(theta|G)} [y] | x; theta) if I understand correctly. It doesn’t make sense to take the expectation (wrt P(theta|G)) of y|x;theta, the conditioning is on the entropy and not inside of the expectation. Theorem 1 - I am not sure if Part 2. b) benefits from the approximate notation that the authors use. I looked in the Supplementary material and the authors derive these relations asymptotically. The asymptotic results may be more beneficial and clearer than how they are stated in Theorem 1.

Relation to Prior Work: Yes prior work is very well presented.

Reproducibility: Yes

Additional Feedback: The authors did a great job!

[Author Response · NeurIPS 2020]

We thank all reviewers for their valuable comments. We addressed your major comments as below:

**[Reviewer 2]: Q1: Provide the rationales about the choice of Dirichlet dis-**
**tribution**. Our proposed framework is designed to predict the subjective opinions
about the classification of testing nodes, such that a variety of uncertainty types,
such as vacuity and dissonance, can be quantified based on the estimated sub-
jective opinions. As a subjective opinion can be equivalently represented by
a Dirichlet distribution about the class probabilities (See Sections 3.1-3.2), we

Figure 1: An illustration of high dissonance and high vacuity based on Dirichlet distribution.

proposed a way to predict the node-level subjective opinions in the form of node-
level Dirichlet distributions. Figure 1 illustrates the effect of Dirichlet distribution for distinguishing each uncertainty
type from others. There are two scenarios: The left scenario has 10 observations for each of the three classes while the
right scenario has 1 observation for each class. The left scenario has a high dissonance while the right scenario has a
high vacuity. These two uncertainty types can be distinguished by using the estimated Dirichlet distributions ($\alpha$ refers
to the Dirichlet parameters), which is not possible by using the estimated class probabilities ($\mathbf{p}$).

**Q2: Clarify the significance and novelty of the contribution**. The significance and novelty of our proposed work lies
in the following unique contributions: (1) Our work is the first that developed the multi-source uncertainty quantification
framework that estimates various types of uncertainties by taking a hybrid approach from both DL and evidence/belief
theory domains for graph data. (2) We provided the first theoretical basis by demonstrating the mathematical proof
that clarifies the relationships between the four important uncertainty types: vacuity, dissonance, aleatoric, and
epistemic uncertainty. (3) We proposed the first-known graph-based Kernel Dirichlet distribution Estimation (GKDE)
approach that estimates node-level Dirichlet distributions based on graph-structural information. We validated its
performance via a theoretical analysis of GKDE as shown in Proposition 1 and an empirical experiment to demonstrate
an extensive performance analysis. We proved the theoretical relationships between the four uncertainty types (the
second contribution) via the mathematical proof and developed the GKDE approach (the third contribution) to support
our proposed multi-source uncertainty quantification framework for graph data (the first contribution).

**[Reviewer 3]: Q1: Discuss and differentiate the different approaches for uncertainty estimation in graphs**.
Existing approaches for graph data have focused on estimating an overall predictive uncertainty for node classification
based on the entropy of the predicted class probabilities for each test node. Some of the recent methods have explicitly
modeled the uncertainty of graph structure in order to better predict the overall predictive uncertainty, such as the
methods based on edge-level dropouts [DropEdge in Rong et. al, 2019], graph Gaussian processes (GPs) [13], and
Bayesian GNNs + MMSBM (mixed membership stochastic block model) [23]. However, no prior work has explored the
decomposition of overall uncertainty into the multiple dimensions as considered in our paper for GNNs, which require
the prediction of second-order uncertainty information about the class probabilities based on Dirichlet distribution. Our
proposed framework can be readily extended to support prediction of the additional uncertainty dimension on graph
structure by integrating graph GPs or MMSBM into our proposed framework.

**Q2: Investigate the performance change under varying a number of labeled**
**nodes.** An empirical analysis of the performance change for misclassification and
OOD detection is shown in Fig 2. The results on the Cora and Citeseer datasets
demonstrate that our proposed method (S-BGCN-T-K) consistently outperformed
the four competitive methods in terms of AUROC and AUPR for varying numbers
of labeled nodes. We also observed worse performance in the AUROC and AUPR
of all the methods under a lower number of labeled nodes. This makes sense as
the less labels we have the higher uncertainty to train the models, resulting in
higher the misclassification rate in overall. **Q3: Clarify the additional KL term**
**for theta.** [5] showed that minimizing the KL term can be well approximated by
minimizing the cross entropy (or squared error) loss function (See Appendix B.8
for detail). Therefore, we used the squared error loss function instead of the KL
term. **Q4: Compare with a (Bayesian) GCN baseline Dropout+DropEdge in**
**Rong et. al, 2019.** As shown in the table below, our proposed method performed better than Dropout+DropEdge on the
Cora and Citeer datasets for misclassificaiton detection. A similar trend was observed for OOD detection.

(a) AUROC on Cora    (b) AURP on Cora

(c) AUROC on Citeseer    (d) AUPR on Citeseer

Figure 2: Performance change with different number of labeled nodes.

**[Reviewer 4]: Q1: Clarify notations**. (1) The semicolon in Equation 5 is a typo and should be replaced by a comma.
(2) About the entropy term in Equation 5, '$P$' is missing before '$(y|x;\boldsymbol{\theta})$' and the the correct entropy term should
be $\mathcal{H}\big[\mathbb{E}_{P(\boldsymbol{\theta}|\mathcal{G})}[P(y|x,\boldsymbol{\theta})]\big]$, referring to the entropy of expected distribution. (3) We agree with you that asymptotic
complexity is a more meaningful metric to represent the efficiency of an algorithm, so we will use it in a revised paper.

| Dataset | Model | AUROC | | | | | AUPR | | | | |
|---------|-------|------|------|------|------|------|------|------|------|------|------|
| | | Va. | Dis. | Al. | Ep. | En. | Va. | Dis. | Al. | Ep. | En. |
| Cora | S-BGCN-T-K | 70.6 | **82.4** | 75.3 | 68.8 | 77.7 | 90.3 | **95.4** | 92.4 | 87.8 | 93.4 |
| | DropEdge | - | - | 76.6 | 56.1 | 76.6 | - | - | 93.2 | 85.4 | 93.2 |
| Citeseer | S-BGCN-T-K | 65.4 | **74.0** | 67.2 | 60.7 | 70.0 | 79.8 | **85.6** | 82.2 | 75.2 | 83.5 |
| | DropEdge | - | - | 71.1 | 51.2 | 71.1 | - | - | 84.0 | 70.3 | 84.0 |

Va.: Vacuity, Dis.: Dissonance, Al.: Aleatoric, Ep.: Epistemic, En.: Entropy

[Meta-Review · NeurIPS 2020]

R#2 and R#3 generally liked the paper. R#1 has a brief review that raised concern on novelty of the method. The rebuttal well addressed the concerns and made all reviewers increase their score. We have collected comments from an additional reviewer, who pointed out more issues on writing and the theoretical results (see blew). We advise the authors to take efforts to address these issues in the revision. ==== Pros: The work proposed a new approach of uncertainty-aware semi-supervised learning on graph data; empirical results are promising. Cons: 1) The method is complicated, putting many different existing methods together in a rather engineering way. 2) The presentation and clarity need to be improved significantly, especially in stating mathematical results. For example, in Theorem 1, u_diss, u_alea, u_epis, u_en were never explicitly defined. The reader need to guess what they are from the section. Unfortunately, the notation does not seem to be consistent across different sessions (e.g., it is unclear how the P(y|p) in Section 3.2 and P(y|x, \theta) in Section 3.3 are related). Because of this, Theorem 1 is not self-contain and it is unclear what are the rigorous mathematical conditions are needed. The use of notation "\gg" and "\approx" should be avoided in stating rigorous mathematical results. It should be stated clearly in what sense the approximation holds. 3) It is unclear if the proposed quantities (when estimated from data) do measure the promised types of uncertainties on the corresponding names. In fact, it is unclear if and when they are identifiable from empirical observation (even when there is infinite data). I would like to suggest to rephrase the writing to clarify the heuristic nature of the proposed method and avoid the impression that they are theoretically rigorous quantification of different uncertainties. In addition, Proposition 1 is very informal and no rigorous condition is presented to arrive the results stated. Also, Eq 11 is very confusing. It is unclear how the second and third terms depend on theta, especially given that theta has been integrated out in P(y|A, r; G) as shown in Eq 8. If variational inference is used somehow, should not we optimize the distribution parameters of theta, instead of theta itself? Misc: y_{ij} in Eq 9 is inconsistent with y_i appeared earlier. Line 107: if a_k = 1/K, why does it still appear in Eq (2)? Eq 5: unclear if the LHS is conditioned on x. Line 143: how is OOD related to alpha=1,1,1? Line 181: is Prob() the same as P()? Section 5.2: theta and f_i are undefined when they first appeared; is the theta here the same as the theta in Section Section 3.3? It is unclear how the variational inference objective is combined Eq 9 and Eq 11; specify it explicitly in the paper.